# Choline supplementation in early life improves and low levels of choline can impair outcomes in a mouse model of Alzheimer's disease

**Elissavet Chartampila[1†], Karim S Elayouby[1‡], Paige Leary[1,2], John J LaFrancois[1,3], David Alcantara-Gonzalez[1,3], Swati Jain[1], Kasey Gerencer[1§], Justin J Botterill[1#], Stephen D Ginsberg[1,2,4,5], Helen E Scharfman[1,2,3,4,5]***

[1]Center for Dementia Research, The Nathan Kline Institute for Psychiatric Research, Orangeburg, United States; [2]Department of Neuroscience and Physiology, New York University Grossman School of Medicine, New York, United States; [3]Departments of Child and Adolescent Psychiatry, New York University Grossman School of Medicine, New York, United States; [4]Department of Psychiatry, New York University Grossman School of Medicine, New York, United States; [5]NYU Neuroscience Institute, New York University Grossman School of Medicine, New York, United States

**\*For correspondence:**
Helen.Scharfman@nki.rfmh.org

**Present address:** †Department of Cell Biology and Physiology, University of North Carolina, Chapel Hill, United States; ‡Department of Neurology, Mount Sinai School of Medicine, New York, United States; §Department of Psychology, University of Maine, Orono, United States; #Department of Anatomy, Physiology, & Pharmacology, College of Medicine, Saskatoon, Canada

**Abstract** Maternal choline supplementation (MCS) improves cognition in Alzheimer's disease (AD) models. However, the effects of MCS on neuronal hyperexcitability in AD are unknown. We investigated the effects of MCS in a well-established mouse model of AD with hyperexcitability, the Tg2576 mouse. The most common type of hyperexcitability in Tg2576 mice are generalized EEG spikes (interictal spikes [IIS]). IIS also are common in other mouse models and occur in AD patients. In mouse models, hyperexcitability is also reflected by elevated expression of the transcription factor ΔFosB in the granule cells (GCs) of the dentate gyrus (DG), which are the principal cell type. Therefore, we studied ΔFosB expression in GCs. We also studied the neuronal marker NeuN within hilar neurons of the DG because reduced NeuN protein expression is a sign of oxidative stress or other pathology. This is potentially important because hilar neurons regulate GC excitability. Tg2576 breeding pairs received a diet with a relatively low, intermediate, or high concentration of choline. After weaning, all mice received the intermediate diet. In offspring of mice fed the high choline diet, IIS frequency declined, GC ΔFosB expression was reduced, and hilar NeuN expression was restored. Using the novel object location task, spatial memory improved. In contrast, offspring exposed to the relatively low choline diet had several adverse effects, such as increased mortality. They had the weakest hilar NeuN immunoreactivity and greatest GC ΔFosB protein expression. However, their IIS frequency was low, which was surprising. The results provide new evidence that a diet high in choline in early life can improve outcomes in a mouse model of AD, and relatively low choline can have mixed effects. This is the first study showing that dietary choline can regulate hyperexcitability, hilar neurons, ΔFosB, and spatial memory in an animal model of AD.

## eLife assessment

In this **fundamental** work, the authors demonstrated that maternal choline supplementation improved spatial memory, reduced hyperexcitability, and restored NeuN expression in a familial Alzheimer's disease mouse model. Interestingly, choline deficiency increased mortality, while paradoxically reduced hyperexcitability. Through behavioral, electrophysiological, and histological

measures, the authors present **convincing** evidence supporting the significant role of maternal choline supplementation in protecting hippocampal functions vulnerable to Alzheimer's disease.

## Introduction

Diet has been suggested to influence several aspects of brain health. One dietary intervention that has been studied extensively in rodents and humans is supplementation of the maternal diet with the nutrient choline (maternal choline supplementation [MCS]). MCS improves many aspects of brain health in humans (*Zeisel and da Costa, 2009*; *Jiang et al., 2014*). In normal rats, MCS also is beneficial, with numerous studies showing improved behavior in the adult offspring (for review, see *Meck and Williams, 2003*).

In Alzheimer's disease (AD), changes to the diet have been recommended (*Bourre, 2006*; *Power et al., 2019*; *Mao et al., 2021*; *Lobo et al., 2022*), including MCS (*Strupp et al., 2016*; *Velazquez et al., 2020*; *Dave et al., 2023*; *Judd et al., 2023a*; *Judd et al., 2023b*). One reason for the recommendation is that serum levels of choline are low in individuals with AD (*Dave et al., 2023*; *Judd et al., 2023b*). In addition, using mouse models of AD, increased choline improved many characteristics of the disease, ranging from inflammation to glucose metabolism and the hallmark amyloid and tau pathology (*Dave et al., 2023*; *Judd et al., 2023a*; *Judd et al., 2023b*). In the Ts65Dn mouse model of Down syndrome (DS) and AD, MCS led to improved memory and attention in Ts65Dn offspring (*Strupp et al., 2016*; *Powers et al., 2017*; *Powers et al., 2021*). In addition, degeneration of basal forebrain cholinergic neurons (BFCNs) in Ts65Dn mice, a hallmark of DS and AD, was reduced (*Kelley et al., 2016*; *Powers et al., 2017*; *Alldred et al., 2023*; *Gautier et al., 2023*).

Here, we asked whether MCS would improve Tg2576 mice, a model of familial AD where a mutation in the precursor to amyloid β (Aβ), amyloid precursor protein (APP), is expressed by the hamster prion protein promoter (*Hsiao et al., 1996*). This is a commonly used mouse model that simulates aspects of AD.

One of the reasons to use Tg2576 mice was to ask whether MCS would improve the hyperexcitability found in the mice. The Tg2576 mouse is ideal because hyperexcitability is robust (*Bezzina et al., 2015*; *Kam et al., 2016*; *Lisgaras and Scharfman, 2023*), and BFCNs are likely to play a role in the hyperexcibililty (*Kam et al., 2016*; *Lisgaras and Scharfman, 2023*). In our past work, the primary type of hyperexcitability was interictal spikes (IIS), which are named because they occur in between seizures (ictal events) in epilepsy. They are studied by EEG. Importantly, IIS are found in numerous mouse models of AD (J20 [*Brown et al., 2018*]; Ts65Dn *Presenilin2*-/- [*Lisgaras and Scharfman, 2023*]; amyloid precursor protein/Presenilin1 [*Shoob et al., 2023*]), as well as patients (*Sanchez et al., 2012*; *Vossel et al., 2016*; *Beagle et al., 2017*; *Vossel et al., 2017*; *Vossel and Karageorgiou, 2021*; *Vossel, 2023*).

We also studied hyperexcitability reflected by high protein expression of ΔFosB, a transcription factor that is increased when neurons have been highly active over the prior 10–14 days (*McClung et al., 2004*). We studied ΔFosB in granule cells (GCs) of the dentate gyrus (DG) because numerous ΔFosB-expressing GCs occur when there are IIS (*You et al., 2017*; *You et al., 2018*). We also focused on GCs because we found that the cell layer of GCs (GCL) is where IIS are largest relative to area CA1 and overlying neocortex (*Lisgaras and Scharfman, 2023*). Moreover, closed-loop optogenetic silencing of GCs reduced IIS (*Lisgaras and Scharfman, 2023*).

We also asked whether behavior would improve if MCS reduced IIS frequency in Tg2576 mice. The basis for this question is in studies of epilepsy, where IIS disrupt cognition (*Rausch et al., 1978*; *Aarts et al., 1984*; *Holmes and Lenck-Santini, 2006*; *Kleen et al., 2010*; *Kleen et al., 2013*; *Gelinas et al., 2016*). Also, both IIS and longer-lasting epileptiform discharges, interictal epileptiform discharges (IEDs), are increased when cognitive dysfunction is impaired in AD patients (*Vossel et al., 2016*).

We fed dams one of three diets, which were relatively low, intermediate, or high in choline (*Figure 1A*; *Supplementary file 1*). The high choline diet provided levels of choline similar to other studies of MCS in rodents (*Meck et al., 1988*; *Loy et al., 1991*; *Holler et al., 1996*; *Meck and Williams, 1999*; *Sandstrom et al., 2002*; *Mellott et al., 2004*; *Glenn et al., 2012*; *Kelley et al., 2019*). After weaning, the intermediate diet was used. Offspring were implanted with electrodes for EEG at 1 month of age, and 24 hr-long recordings were made every month shortly thereafter and ending at 6 months of age.

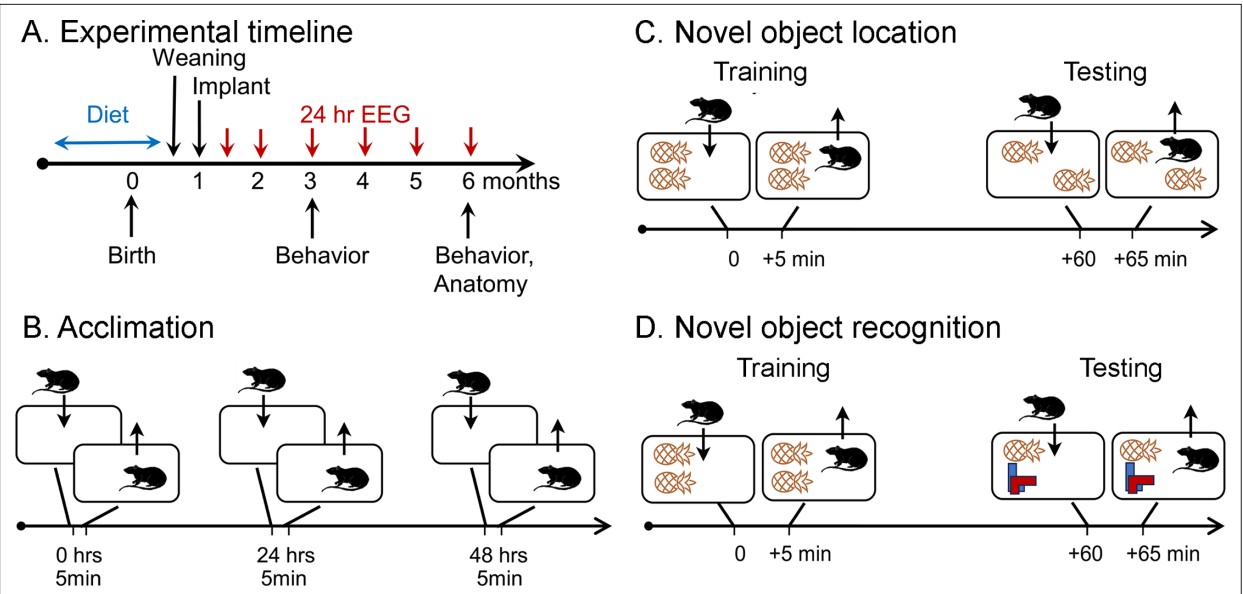

**Figure 1.** Schematic of the behavioral procedures for the novel object location (NOL) and novel object recognition (NOR) tasks. (**A**) Experimental timeline. Dams were fed one of three diets for mating, gestation, and offspring that consumed the diet until weaning. At 1 month old, offspring were implanted with electrodes for EEG. Recordings for 24 hr started 1 week later (5 weeks old; 1.2 months old in the graphs). Additional recordings were made at 2, 3, 4, 5, and 6 months of age. At 3 and 6 months of age, behavior was tested, mice were perfused, brains were sectioned, and sections were processed with antibodies against NeuN and ΔFosB. (**B**) Prior to NOL or NOR, animals were acclimated to the testing arena. There were three acclimation sessions separated by 24 hr during which animals were allowed to freely explore for 5 min. (**C**) In the NOL task, animals were placed in a cage with two identical objects and allowed to freely explore for 5 min (Training). After 1 hr, they were brought back to the cage, where one object was displaced, and allowed to freely explore for 5 min (Testing). (**D**) In the NOR task. animals were placed in a cage with two identical objects and allowed to freely explore for 5 min (Training). After 1 hr, they were brought back to the cage, where one object was replaced, and allowed to freely explore for 5 min (Testing).

The online version of this article includes the following figure supplement(s) for figure 1:

**Figure supplement 1.** Details of behavioral tasks.

The time span (1–6 months of age) is relevant because IIS are present at these ages (**Bezzina et al., 2015**; **Kam et al., 2016**; **Lisgaras and Scharfman, 2023**). The novel object location (NOL) task is impaired also (at 3–4 months of age in Tg2576 mice [**Duffy et al., 2015**]), so we evaluated NOL at 3 months. We also tested a related task, novel object recognition (NOR [**Vogel-Ciernia and Wood, 2014**]). To understand the persistence of any effects of the diets, we repeated NOL and NOR at 6 months. The interval between testing was sufficiently long that it is unlikely that testing at 3 months affected testing at 6 months, but we cannot exclude the possibility. At 6 months of age, we perfusion-fixed mice, sectioned the brains, and evaluated NeuN and ΔFosB protein expression. NeuN was studied because it is a neuronal marker that is reduced in numerous pathological conditions (**Buckingham et al., 2008**; **Kadriu et al., 2009**; **Matsuda et al., 2009**; **Duffy et al., 2011**; **Duffy et al., 2015**). We also assayed ΔFosB because it can be used to assay neuronal activity and therefore is complementary to EEG recordings. It also can be used to assess hyperexcitability of GCs.

The results showed a remarkable effect of the high choline diet. IIS and spatial memory were improved, as was NeuN and ΔFosB expression. Interestingly, the relatively low choline diet had mixed effects, reducing IIS frequency, but making NOL, ΔFosB, and NeuN worse. The mice also died prematurely relative to mice that were fed the high choline diet. We also report for the first time that there is loss of NeuN-ir in the DG hilus of Tg2576 mice, which is important because abnormal hilar neurons could cause GC hyperexcitability (**Sperk et al., 2007**; **Scharfman and Myers, 2012**). In summary, we make a strong argument for choline supplementation in early life to improve outcomes in an AD model, especially the DG.

# Results

## Approach

As shown in *Figure 1A*, animals were implanted with electrodes at 1 month of age and recorded for a continuous 24 hr-long period at 5 weeks, 2, 3, 4, 5, and 6 months of age. At 3 and 6 months of age, mice were tested for NOL and NOR. Afterward, they were perfused and immunocytochemistry was conducted for NeuN and ΔFosB. Sections were made in the coronal plane and relatively anterior and posterior levels were compared to sample different parts of hippocampus. Some mice were not possible to include in all assays either because they died before reaching 6 months or for other reasons.

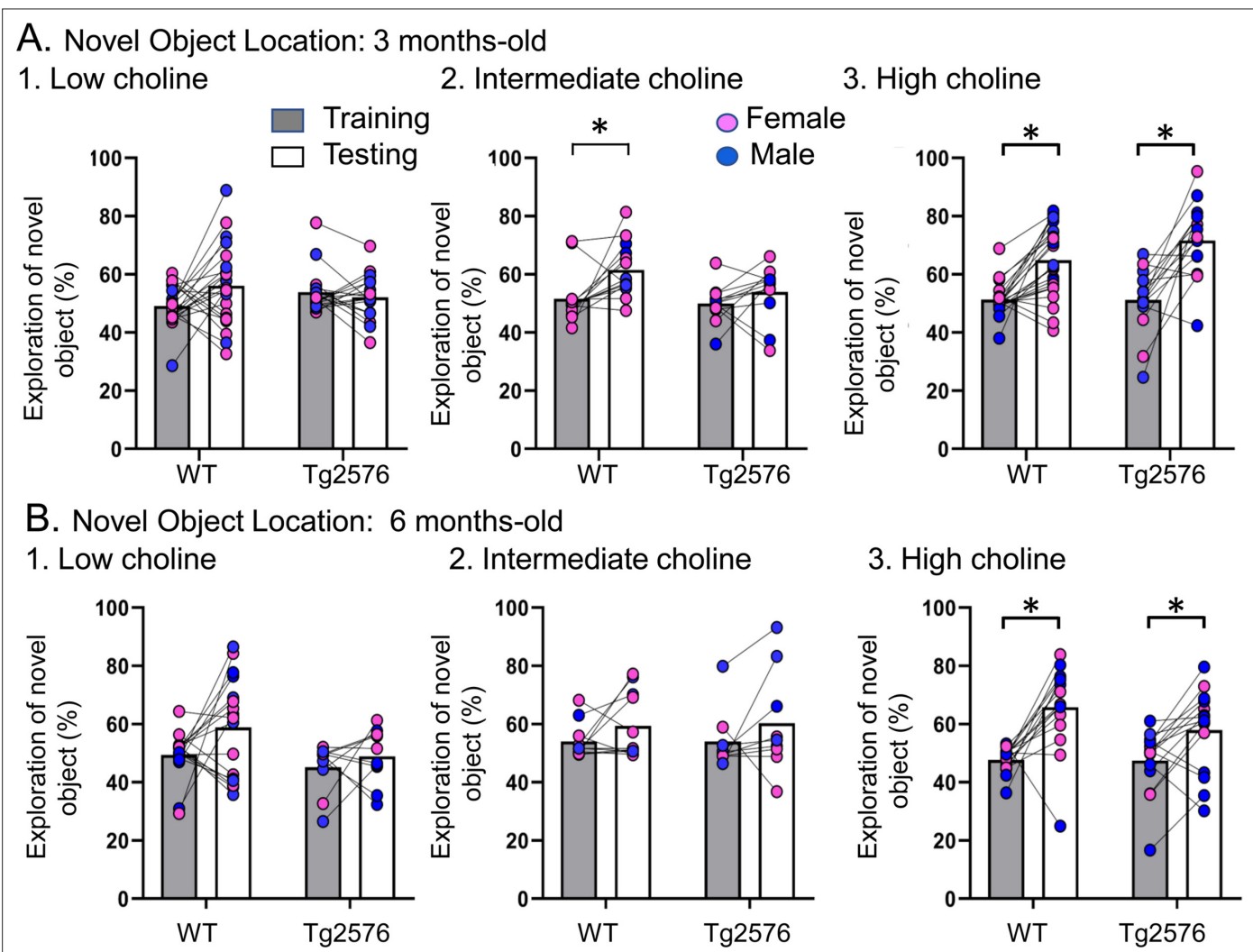

**Figure 2.** Choline enrichment reduced spatial memory deficits. (**A**) Three months-old mice. For (**A**) and (**B**), breeders were fed the specified diet and offspring were tested. 1. Low choline diet. WT and Tg2576 offspring showed spatial memory deficits. 2. Intermediate diet. Tg2576 showed spatial memory deficits but not WT. 3. High choline diet. Spatial memory was improved. (**B**) Six months-old mice. 1. Low choline diet. WT and Tg2576 offspring showed spatial memory deficits. 2. Intermediate diet. WT and Tg2576 mice showed spatial memory deficits. 3. High choline diet. WT and Tg2576 mice had improved spatial memory.

The online version of this article includes the following figure supplement(s) for figure 2:

**Figure supplement 1.** The data shown in *Figure 2* are plotted with means ± sem.

**Figure supplement 2.** Mortality was high in mice treated with the low choline diet.

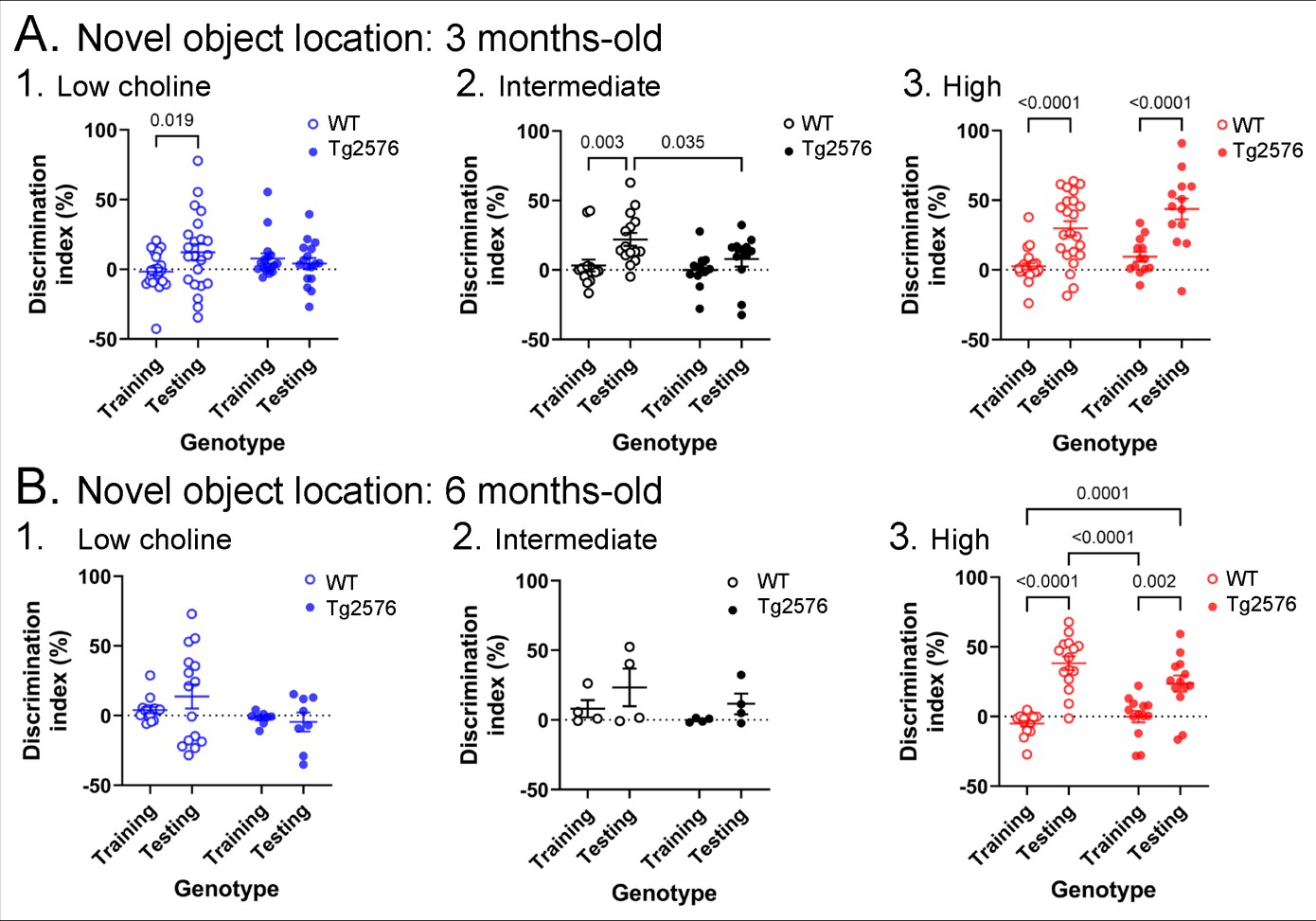

**Figure 3.** Novel object location results based on the discrimination index. (**A**) Results are shown for 3 months-old WT and Tg2576 mice based on the discrimination index. 1. Mice fed the low choline diet showed object location memory only in WT. 2. Mice fed the intermediate diet showed object location memory only in WT. 3. Mice fed the high choline diet showed memory both for WT and Tg2576 mice. Therefore, the high choline diet improved memory in Tg2576 mice. (**B**) The results for 6 months-old mice are shown. 1–2. There was no significant diference between training and testing demonstrated by mice that were fed the low or intermediate choline diet. 3. Mice fed a diet enriched in choline showed significant differences between training and testing whether they were WT or Tg2576 mice. Therefore, choline enrichment improved task performance in all mice.

### Behavior

NOL and the NOR were selected because they show deficits in Tg2576 mice at just 3 months of age (*Duffy et al., 2015*). This is before deficits have been shown in other tasks (i.e., Morris Water Maze, Radial Arm Water Maze, Y-Maze) which occur after 9 months of age (*Yassine et al., 2013*; *Wolf et al., 2016*). *Figure 1B–D* (see also *Figure 1—figure supplement 1*) shows a schematic of the specific experimental procedures for NOL and NOR.

### Novel object location

The results of the NOL task are presented in *Figure 2*, *Figure 3*, *Figure 4*. In *Figure 2*, lines connect the data for a given animal in training and testing. Data are the time spent exploring the objects. Exploration of the object that is moved in the training session (the novel object) is expressed as a percentage of the total exploration time of both objects. When the slope of the line increases between training and testing, the time spent exploring the moved object increased in testing relative to training. This increase reflects a preference to explore the moved object during the testing period and suggests the mouse can recall the old object locations and can recognize the new object location. *Figure 2—figure supplement 1* shows the data with mean ± sem.

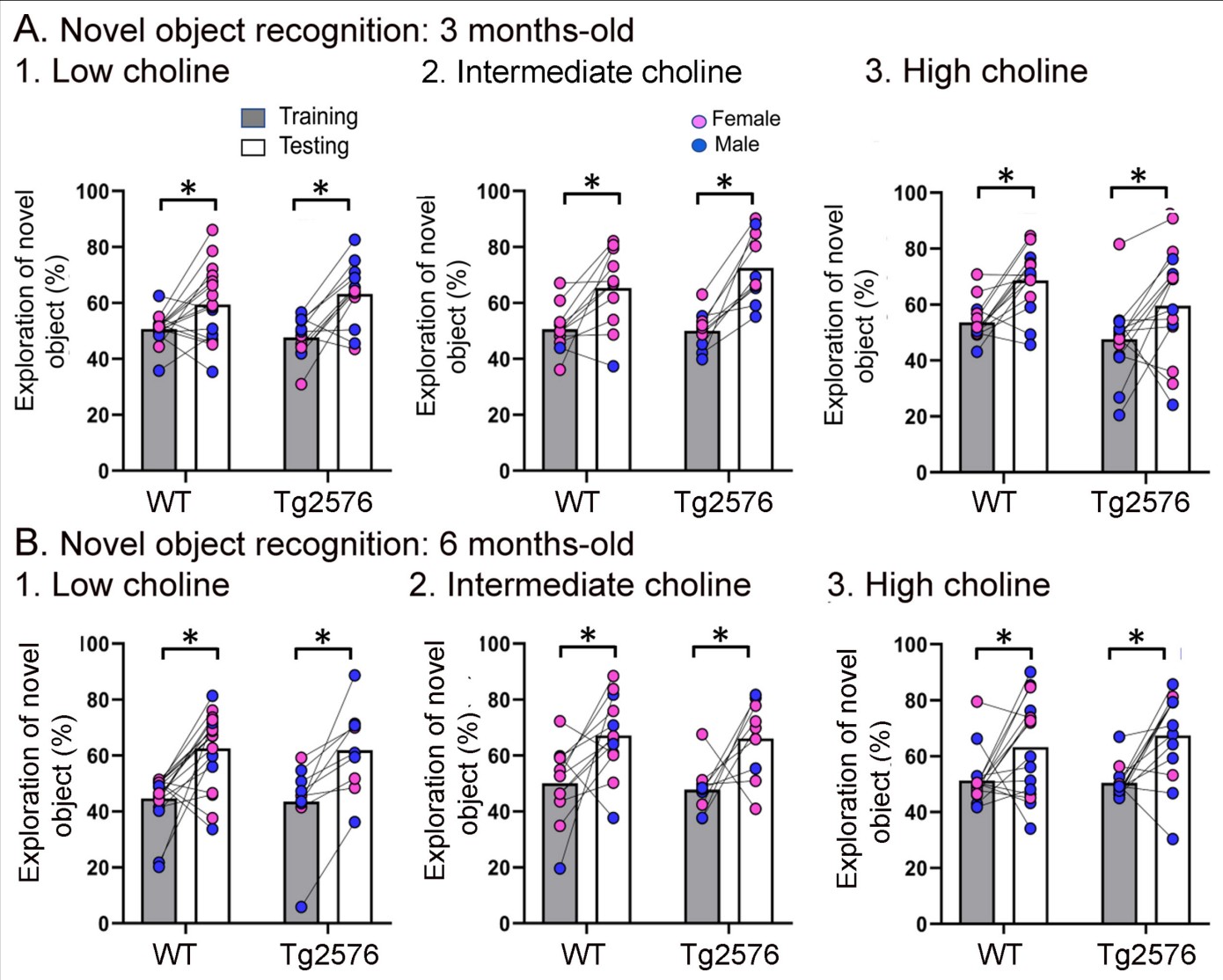

**Figure 4.** WT and Tg2576 mice showed object recognition memory regardless of diet. (**A**) Three months-old mice. 1–3. WT and Tg2576 mice performed the novel object recognition (NOR) task. (**B**) Six months-old mice. 1–3. WT and Tg2576 mice performed the NOR task.

The online version of this article includes the following figure supplement(s) for figure 4:

**Figure supplement 1.** The data shown in *Figure 3* are plotted with means ± sem.

An important foundation for this task is equal preference for the objects during training. The lack of preference reflects no inherent bias for one object vs. the other. We confirmed the lack of bias for each genotype and each diet during training. To conduct statistics, we first conducted a two-way ANOVA based on the training data with genotype or age (3 or 6 months) as factors. There were no effects of genotype ($F_{(1,97)} = 0.40$, $p=0.530$) or age ($F_{(1,70)} = 1.94$, $p=0.167$) on object exploration time. Then, we conducted a two-way ANOVA to ask whether diet played a role. The main factors were diet and age, and there also was no effect of diet ($F_{(2,97)} = 0.06$, $p=0.941$) or age ($F_{(2,70)} = 2.94$, $p=0.058$) on object exploration time. Thus, novel object exploration during training approached 50% in all treatment groups, independent of genotype and maternal diet, and at both ages.

Next, we confirmed that Tg2576 mice were deficient in the test phase of NOL when they were fed the standard mouse diet (the intermediate diet). First, we studied 3 months-old mice (*Figure 2A*). We used a two-way ANOVA with genotype and task phase (training vs. testing) as main factors. There was a significant main effect of the task phase ($F_{(1,48)} = 8.51$, $p=0.005$) and a trend for a genotype effect ($F_{(1,48)} = 3.80$, $p=0.057$). Consistent with previous studies (*Duffy et al., 2015*), Tukey–Kramer post

hoc analyses revealed a significant increase in the exploration of the novel object during testing in WT (p=0.017) but not Tg2576 mice (p=0.676; *Figure 2A*). Thus, Tg2576 mice were impaired in NOL when fed the intermediate diet.

In contrast to Tg2576 mice fed the intermediate diet, Tg2576 mice that received the high choline diet showed memory for object location. Thus, a two-way ANOVA revealed a main effect of the task phase (F(1,72) = 46.16, p<0.0001) but not genotype (F(1,72)=1.74, p=0.191). Tukey-Kramer post hoc analyses revealed a significant increase in the exploration of the novel object during testing in both WT (p=0.0003) and Tg2576 mice (p<0.0001; *Figure 2A3*). Therefore, high choline supplementation improved memory in Tg2576 mice.

In animals treated with the low choline diet, there was no effect of the phase of the task (F(1,74) = 1.41, p=0.250) or genotype (F(1,74) = 0.027, p=0.871). These data suggest that relatively low levels of choline during early life impaired spatial memory in both WT and Tg2576 mice. Thus, low choline had a significant adverse effect. Adverse effects are further supported by survival plots showing that there was more mortality at earlier ages in offspring exposed to the low choline diet (*Figure 2—figure supplement 2*).

*Figure 2B* shows the results in 6 months-old WT and Tg2576 mice. In animals that received the intermediate diet, a two-way ANOVA showed no effect of genotype (F(1,36) = 0.01, p=0.907) or task phase (F(1,36) = 2.36, p=0.133), revealing memory deficits in WT and Tg2576 mice (*Figure 2B*). In contrast, the high choline group showed a main effect of task phase (F(1,56) = 22.18, p<0.0001) but not genotype (F(1,56) = 1.78, p=0.188). Tukey–Kramer post hoc analyses showed a significant increase in the exploration of the novel object during testing in both WT (p<0.001) and Tg2576 (p=0.020; *Figure 2B*). Thus, the high choline-treated mice showed object location memory but the mice fed the intermediate diet did not. The mice that received the relatively low choline diet showed an effect of genotype (F(1,50) = 4.36, p=0.042; *Figure 2B*) in that training in WT mice differed from testing in Tg2576 mice (p=0.031). However, there was no effect of task phase (F(1,50) = 3.75, p=0.058). Thus, WT mice did not differ between training and testing (p=0.114) and Tg2576 mice did not either (p=0.921). Therefore, mice fed the low choline and intermediate diets were impaired and the high choline-treated mice were not.

The discrimination indices are shown in *Figure 3* and results led to the same conclusions as the analyses in *Figure 2*. For the 3 months-old mice (*Figure 3A*), the low choline group did not show the ability to perform the task for WT or Tg2576 mice. Thus, a two-way ANOVA showed no effect of genotype (F(1,74) = 0.027, p=0.870) or task phase (F(1,74) = 1.41, p=0.239). For the intermediate diet-treated mice, there was no effect of genotype (F(1,50) = 0.52, p=0.067) but there was an effect of task phase (F(1,50) = 8.33, p=0.006). WT mice showed a greater discrimination index during testing relative to training (p=0.019) but Tg2576 mice did not (p=0.664). Therefore, Tg2576 mice fed the intermediate diet were impaired. In contrast, high choline-treated mice performed well. There was a main effect of task phase (F(1,68) = 39.61, p=<0.001) with WT (p<0.0001) and Tg2576 mice (p=0.0002) showing preference for the moved object in the test phase. Interestingly, there was a main effect of genotype (F(1,68) = 4.50, p=0.038) because the discrimination index for WT training was significantly different from Tg2576 testing (p<0.0001) and Tg2576 training was significantly different from WT testing (p=0.0003).

The discrimination indices of 6 months-old mice led to the same conclusions as the results in *Figure 2*. There was no evidence of discrimination in low choline-treated mice by two-way ANOVA (no effect of genotype, F(1,42) = 3.25, p=0.079; no effect of task phase, F(1,42) = 0.28, p=0.601). The same was true of mice fed the intermediate diet (genotype, F(1,12) = 1.44, p=0.253; task phase, F(1,12) = 2.64, p=0.130). However, both WT and Tg2576 mice performed well after being fed the high choline diet (effect of task phase, F(1,52) = 58.75, p=0.0001, but not genotype, F(1,52) = 1.20, p=0.279). Tukey–Kramer post hoc tests showed that both WT (p<0.0001) and Tg2576 mice that had received the high choline diet (p=0.0005) had elevated discrimination indices for the test session.

Taken together, these results demonstrate the lasting beneficial effects of the high choline diet and adverse effects of low choline on offspring in a spatial memory task in Tg2576 as well as WT littermates.

## Novel object recognition

We first confirmed that animals did not show preference for one object over the other in training. Indeed, a two-way ANOVA revealed no main effects of genotype at 3 months ($F_{(1,71)}$ = 2.59, p=0.536) and no effect of diet ($F_{(2,71)}$ = 0.22, p=0.809). The same result was obtained at 6 months of age (genotype, $F_{(1,66)}$ = 0.11, p=0.746; diet, $F_{(2,66)}$ = 0.98, p=0.376). Thus, novel object exploration during training approached 50% in all treatment groups, independent of genotype and maternal diet, and at both ages.

*Figure 4* shows the results for the NOR task. Lines connect the performance of a given animal in training and in testing. *Figure 4—figure supplement 1* shows the data with mean ± sem. *Figure 4A* shows that there were no impairments at 3 months of age. In animals treated with the low choline diet, there was a main effect of task phase ($F_{(1,52)}$ = 19.81, p<0.0001), no effect of genotype ($F_{(1,52)}$ = 0.02, p=0.887), and more exploration of the novel object during the test phase (WT, p=0.007; Tg2576, p<0.002). In animals that received the intermediate diet, there also was a significant main effect of task phase ($F_{(1,38)}$ = 30.88, p<0.0001) and no effect of genotype ($F_{(1,38)}$ = 0.97, p=0.330). Animals explored the novel object significantly more during testing in WT (p=0.014) and Tg2576 mice (p=0.002; *Figure 3B*). For the high choline group, there also was an effect of task phase ($F_{(1,26)}$ = 17.51, p=0.0003) and not genotype ($F_{(1,26)}$ = 3.53, p=0.072) and a significant increase in the exploration of the novel object during testing in both WT (p=0.006) and Tg2576 (p=0.027; *Figure 3B*).

*Figure 4B* shows the data for 6 months of age. Animals that received the low choline diet performed the NOR task at 6 months of age (two-way ANOVA, task phase, $F_{(1,26)}$ = 32.66, p<0.0001; genotype, $F_{(1,26)}$ = 0.048, p=0.821). Tukey–Kramer post hoc tests showed that there was a significant increase in the exploration of the novel object in testing both in WT (p<0.001) and Tg2576 (p=0.003). For mice that had been fed the intermediate choline diet, they also performed the NOR task at 6 months of age (two-way ANOVA, task phase, $F_{(1,19)}$ = 13.65, p=0.002; genotype, $F_{(1,19)}$ = 0.28, p=0.604). Tukey–Kramer post hoc analyses revealed a significant increase in the exploration of the novel object

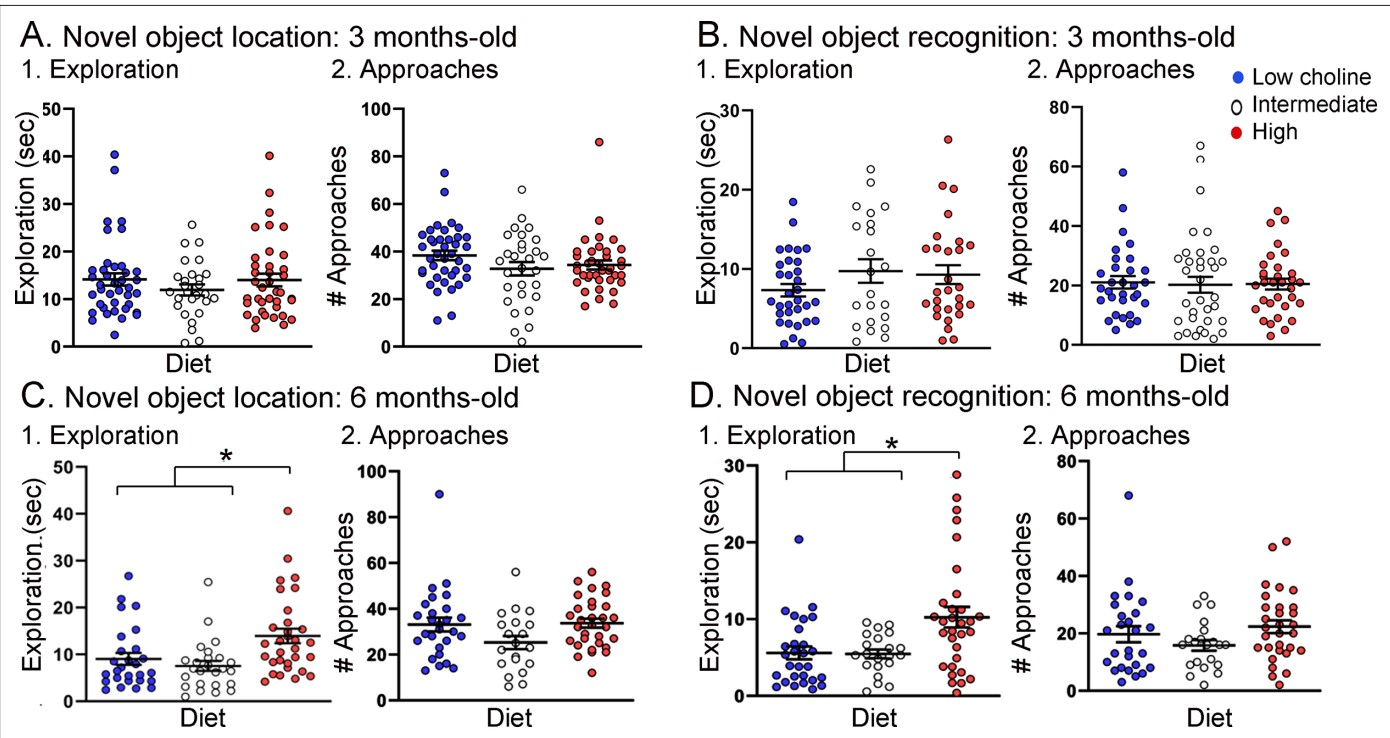

**Figure 5.** Tg2576 mice treated with the high choline diet spent more time with the objects at each approach. (**A, B**) Novel object location (NOL). There was no effect of diet on time exploring or approaches to the novel object in the NOL task at 3 months of age (**A**) but there were significant differences at 6 months of age (**B**). Mice exposed to the high choline diet spent more time exploring than mice that had been fed the low choline or the intermediate diet. (**C, D**) Novel object recognition (NOR). There was no effect of diet on time exploring or approaches to the novel object in the NOR task at 3 months of age (**C**) but there were significant differences at 6 months of age (**D**). Mice that had been fed the high choline diet spent more time exploring than mice that had been fed the low choline or the intermediate diet.

during testing compared to training for both WT (p=0.0027) and Tg2576 (p=0.039; *Figure 4B*). Mice fed the high choline diet also showed no deficit in NOR at 6 months of age. Thus, a two-way ANOVA revealed a main effect of task phase (F(1,27) = 16.26, p=0.0004) but not genotype (F(1,27) = 0.24, p=0.625). Tukey–Kramer post hoc analyses revealed a significant increase in the exploration of the novel object during testing in both WT (p=0.003) and Tg2576 (p=0.007; *Figure 4B*).

Thus, at both 3 and 6 months of age, WT and Tg2576 mice performed well in the NOR task. These results suggest that young Tg2576 mice are less sensitive to NOR than NOL. The greater sensitivity to NOL is consistent with past demonstrations that the DG contributes to NOL (*Sahay et al., 2011*; *Kesner et al., 2015*; *Spyrka and Hess, 2018*; *Vandrey et al., 2020*; *Gulmez Karaca et al., 2021*; *GoodSmith et al., 2022*). Also, our implementation of NOL may have increased the DG dependence of the task by making the object locations relatively close together because in studies by Pofahl and colleagues it was shown that distances between objects like those we used made NOL DG-dependent (*Pofahl et al., 2021*).

## Exploration time

Total object exploration (TOE) was measured to address any effects of genotype or diet on the total duration of exploration of objects (*Figure 5*). For NOL TOE at 3 months of age, there were effects of genotype (F(1,67) = 6.89, p=0.01) but not diet (F(2,87) = 0.67, p=0.63). Tukey–Kramer post hoc tests showed that Tg2576 mice fed the low choline diet showed more exploration than WT mice fed the same diet but the effect was on the border of significance (p=0.049; data not shown). Three months-old mice tested with NOR showed no significant effect of genotype (F(1,76) = 1.35, p=0.25) or diet (F(2,76) = 0.30, p=0.61). Since there were weak or no effects of genotype, all genotypes are pooled for *Figure 5A1* (NOL) and *Figure 5B1* (NOR).

For NOL TOE at 6 months of age, there was no effect of genotype (F(1,66) = 0.33; p=0.57). For NOR TOE, the same was true (genotype: F(1,72) = 0.96, p=0.33; diet: F(2,72) = 8.50, p=0.0005). Because genotype was not a significant factor, we pooled genotypes (NOL, *Figure 5C1*; NOR, *Figure 5D2*).

With genotypes pooled, we determined how diets differed. For NOL TOE, mice treated with the low choline diet had less exploration than mice fed the high choline diet (one-way ANOVA, F(2,77) = 6.90; p=0.002; Tukey post hoc test, p=0.005; *Figure 5C1*) and the mice that were fed the intermediate diet also had less exploration than mice fed the high choline diet (Tukey post hoc test, p=0.008; *Figure 5C1*). Results for NOR were similar: mice treated with the low choline diet had less exploration than mice fed the high choline diet (one-way ANOVA, F(2,74) = 4.81; p=0.020; Tukey post hoc test, p=0.016; *Figure 5D1*) and the mice that were fed the intermediate diet also had less exploration than mice fed the high choline diet (Tukey post hoc test, p=0.03; *Figure 5D1*).

To gain insight into the potential reason for the effect of diet on exploration at 6 months of age, we measured total object approaches (TOA was defined as the number of approaches to the familiar object + number of approaches to the novel object) (*Figure 5*). There was no effect of diet on TOA for NOL (F(2,75) = 2.88; p=0.092; *Figure 5C2*) or NOR (F(2,74) = 1.81, p=0.171; *Figure 5D2*).

Taken together, the results indicate that, in 6 months-old mice, animals that received the high choline diet spent more time with objects at each approach. This could explain the increased object memory in high choline-treated mice because there would be more time for information processing during an approach. Another possibility is that as Tg2576 mice age they show compensatory changes that enhance memory. This has been suggested for procedural learning (*Middei et al., 2004*) but, to the best of our knowledge, it has not been shown for memory of objects. A third possibility is that the high choline diet reduces anxiety (*Glenn et al., 2012*; *Langley et al., 2015*; *McCall et al., 2015*). Reduced anxiety may lessen fear of exploring objects, and as a result, animals may spend more time with objects.

## Anatomy
### NeuN

To further analyze the effects of dietary choline early in life, we used an antibody against a neuronal nuclear antigen, NeuN. Previous studies have found that reduced expression of NeuN often occurs in neurons after insults or injury (e.g., ischemia, toxicity, and even aging; *Lind et al., 2005*; *Portiansky et al., 2006*; *Buckingham et al., 2008*; *Kadriu et al., 2009*; *Matsuda et al., 2009*; *Won et al., 2009*; *Duffy et al., 2011*). For example, when NeuN is reduced after a heterozygous deletion in ankyrin-rich

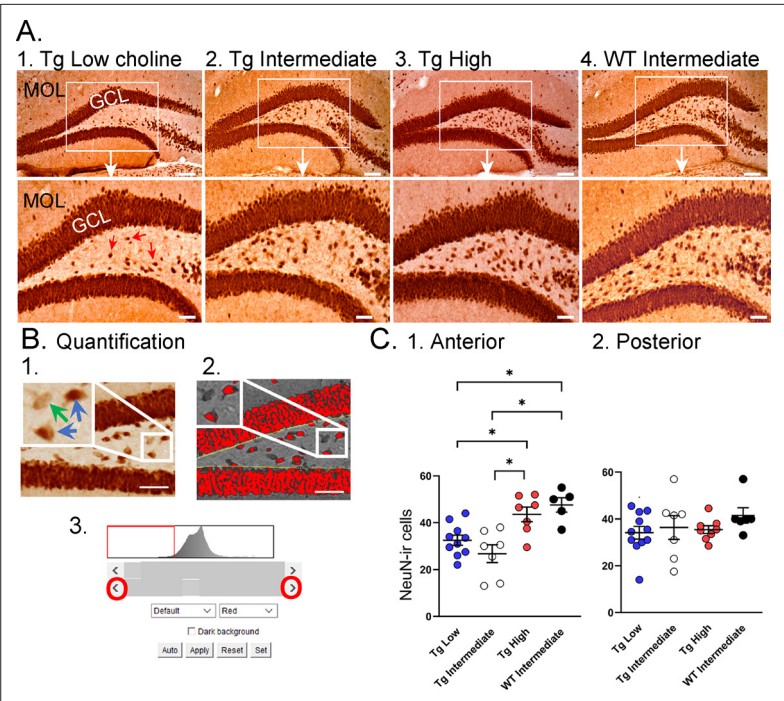

**Figure 6.** Choline supplementation improved NeuN immunoreactivity (ir) in hilar cells in Tg2576 animals. (**A**) Representative images of NeuN-ir staining in the anterior dentate gyrus (DG). 1. A section from a Tg2576 mouse fed the low choline diet. The area surrounded by a box is expanded below. Red arrows point to NeuN-ir hilar cells. Mol = molecular layer; GCL = granule cell layer. Calibration for the top row, 100 μm; for the bottom row, 50 μm. 2. A section from a Tg2576 mouse fed the intermediate diet. Same calibrations as for 1. 3. A section from a Tg2576 mouse fed the high choline diet. Same calibrations as for 1. 4. A section from a WT mouse fed the intermediate diet. Same calibrations as for 1. (**B**) Quantification methods. Representative images demonstrate the thresholding criteria used to quantify NeuN-ir. 1. A NeuN-stained section. The area surrounded by the white box is expanded in the inset (arrow) to show three hilar cells. The two NeuN-ir cells above threshold are marked by blue arrows. The one NeuN-ir cell below threshold is marked by a green arrow. 2. After converting the image to grayscale, the cells above threshold were designated as red. The inset shows that the two cells that were marked by blue arrows are red while the cell below threshold is not. 3. An example of the threshold menu from ImageJ showing the way the threshold was set. Sliders (red circles) were used to move the threshold to the left or right of the histogram of intensity values. The final position of the slider (red arrow) was positioned at the onset of the steep rise of the histogram. (**C**) NeuN-ir in Tg2576 and WT mice. Tg2576 mice had either the low, intermediate, or high choline diet in early life. WT mice were fed the standard diet (intermediate choline). 1. Tg2576 mice treated with the high choline diet had significantly more hilar NeuN-ir cells in the anterior DG compared to Tg2576 mice that had been fed the low choline or intermediate diet. The values for Tg2576 mice that received the high choline diet were not significantly different from WT mice, suggesting that the high choline diet restored NeuN-ir. 2. There was no effect of diet or genotype in the posterior DG.

membrane spanning kinase D-interacting substrate of 220K, ARMS/Kidins, a protein critical to neurotrophin signaling, neurons in the entorhinal cortex demonstrate loss of NeuN immunoreactivity (ir) and the cellular morphology is abnormal (**Duffy et al., 2011**; **Duffy et al., 2015**). NeuN is also reduced when Aβ levels are elevated (**Wu et al., 2016**) and NeuN is reduced in AD patients (**Camporez et al., 2021**). Therefore, we initially examined NeuN-ir in Tg2576 mice. We found it was reduced and then asked if the high choline diet could restore it.

As shown in **Figure 6A**, relatively weak NeuN-ir was observed in Tg2576 that received the low choline diet and the intermediate diet compared to the high choline diet (**Figure 6A**). This observation is consistent with the vulnerability of the hilus to insult and injury (**Scharfman, 1999**). To examine the septotemporal axis, quantification from 2 to 3 anterior coronal sections were averaged to provide a value for the septal pole of the DG and 2–3 posterior sections were averaged to assess the caudal DG. The most ventral, temporal levels were not sampled. There was no effect of diet (F(2,30) = 2.11, p=0.137) nor rostral-caudal level (F(1,30) = 0.02, p=0.877) but there was a significant interaction

of diet and rostral-caudal level (F(2,30) = 3.88, p=0.036). A one-way ANOVA using anterior values showed a significant effect of diet (F(2,21) = 7.58, p=0.003) but this was not the case for posterior values (F(2,23) = 0.13; p=0.876). In the anterior DG, there was less NeuN-ir in mice fed the low choline diet versus the high choline diet (Tukey–Kramer post hoc test, p=0.029) and less NeuN in mice fed the intermediate versus high choline diet (p=0.003). The results suggest that hilar neurons of Tg2576 mice have reduced NeuN-ir and choline enrichment protected dorsal hilar neurons from NeuN loss in Tg2576 mice.

To ask if the improvement in NeuN after the high choline diet in Tg2576 restored NeuN to WT levels, we used WT mice. The WT mice were fed the intermediate diet because it is the standard mouse chow, and this group was intended to reflect normal mice. For the analysis, we used a one-way ANOVA with four groups: low choline Tg2576, intermediate Tg2576, high choline Tg2576, and intermediate WT (*Figure 5C*). Tukey–Kramer multiple comparisons tests were used as the post hoc tests. The results showed a significant group difference for anterior DG (F(3,25) = 9.20; p=0.0003; *Figure 5C1*) but not posterior DG (F(3,28) = 0.867; p=0.450; *Figure 5C2*). Regarding the anterior DG, there were more NeuN-ir cells in high choline-treated mice than both low choline (p=0.046) and intermediate choline-treated Tg2576 mice (p=0.003). WT mice had more NeuN-ir cells than Tg2576 mice fed the low (p=0.011) or intermediate diet (p=0.003). Tg2576 mice that were fed the high choline diet were not significantly different from WT (p=0.827). Thus, we found reduced hilar NeuN in Tg2576 mice and the high choline diet increased NeuN protein expression to WT levels.

## ΔFosB

To complement the information from the video-electroencephalographic recordings (video-EEG) (see below), we used a marker of elevated neuronal activity, ΔFosB (*Figure 7*). ΔFosB is a truncated variant of the transcription factor FosB, which is increased by enhanced neuronal activity; ΔFosB has a half-life of approximately 8 days (*Ulery-Reynolds et al., 2009*), so when ΔFosB is elevated, it reflects increased neuronal activity over the last 10–14 days (*McClung et al., 2004*). Previous studies have shown that when the J20 mouse model of AD is examined with an antibody to ΔFosB, the GC layer shows extremely high levels of ΔFosB expression (*Corbett et al., 2017*), similar to a mouse with chronic spontaneous seizures (*Morris et al., 2000*). This is not surprising since the J20 mice have recurrent seizures (*Palop et al., 2007*). Therefore, we asked if Tg2576 mice would have robust ΔFosB in the GC layer, and choline supplementation would reduce it. We also included WT mice fed the intermediate (standard) diet. These WT mice would allow us to address the possibility that the high choline diet restored ΔFosB back to normal.

There was strong expression of ΔFosB in Tg2576 GCs in mice fed the low choline diet (*Figure 7A1*). The high choline diet and intermediate diet appeared to show less GCL ΔFosB-ir (*Figure 7A2-3*). A two-way ANOVA was conducted with the experimental group (Tg2576 low choline diet, Tg2576 intermediate choline diet, Tg2576 high choline diet, WT intermediate choline diet) and location (anterior or posterior) as main factors. There was a significant effect of group (F(3,32) = 13.80, p=<0.0001) and location (F(1,32) = 8.69, p=0.006). Tukey–Kramer post hoc tests showed that Tg2576 mice fed the low choline diet had significantly greater ΔFosB-ir than Tg2576 mice fed the high choline diet (p=0.0005) and WT mice (p=0.0007). Tg2576 mice fed the low and intermediate diets were not significantly different (p=0.275). Tg2576 mice fed the high choline diet were not significantly different from WT (p>0.999). There were no differences between groups for the posterior DG (all p>0.05).

ΔFosB quantification was repeated with a lower threshold to define ΔFosB-ir GCs (see 'Methods') and results were the same (*Figure 7D*). Two-way ANOVA showed a significant effect of group (F(3,32) = 14.28, p<0.0001) and location (F(1,32) = 7.07, p=0.012) for anterior DG but not posterior DG (*Figure 7D*). For anterior sections, Tukey–Kramer post hoc tests showed that low choline mice had greater ΔFosB-ir than high choline mice (p=0.002) and WT mice (p=0.005) but not Tg2576 mice fed the intermediate diet (p=0.275; *Figure 7D1*). Mice fed the high choline diet were not significantly different from WT (p=0.993; *Figure 7D1*). These data suggest that high choline in the diet early in life can reduce neuronal activity of GCs in offspring later in life. In addition, low choline has an opposite effect, suggesting low choline in early life has adverse effects.

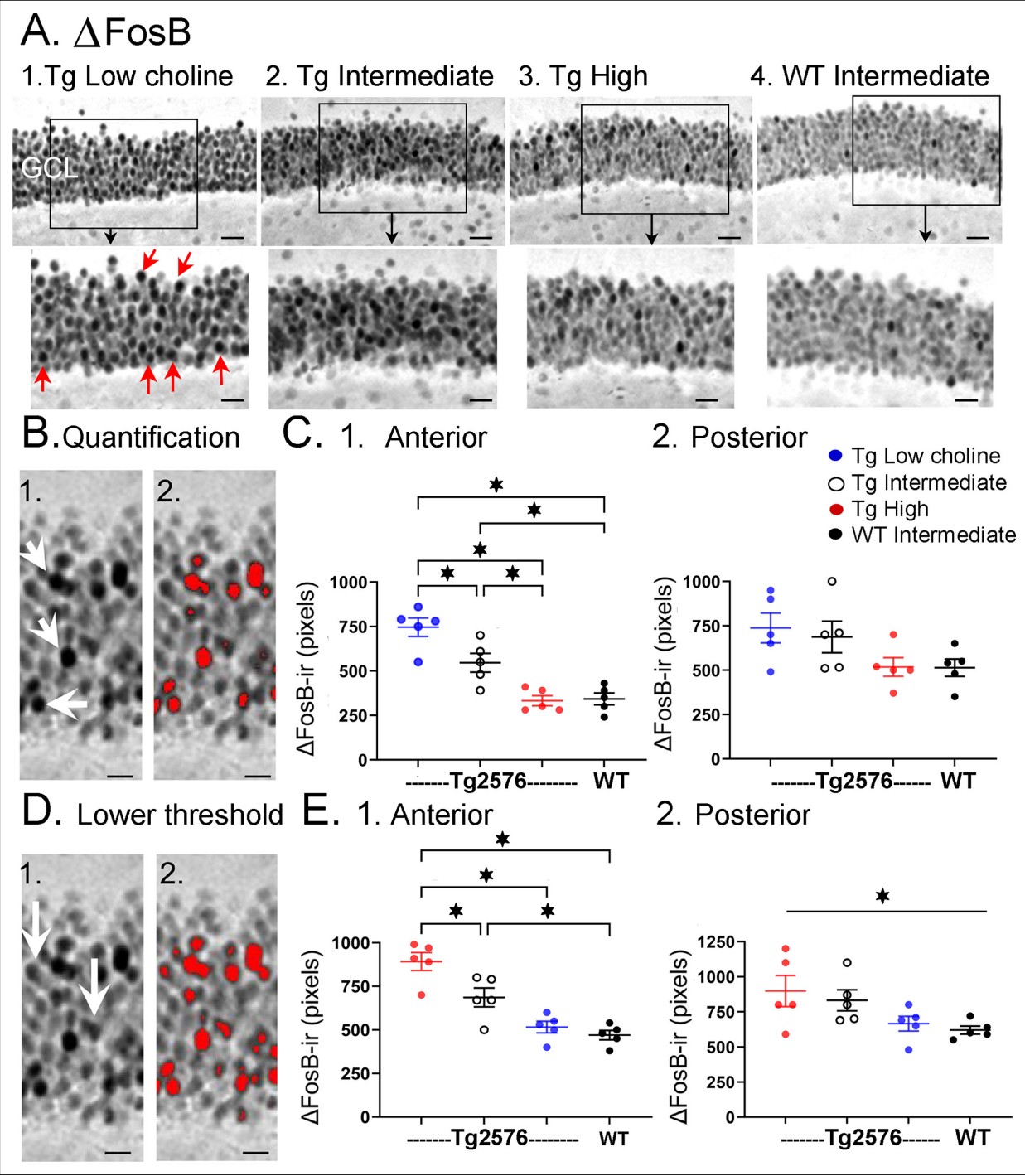

**Figure 7.** Choline supplementation reduced ΔFosB expression in dorsal granule cells (GCs) of Tg2576 mice. (**A**) Representative images of ΔFosB staining in the GCL. A section from a low choline-treated Tg2576 mouse shows robust ΔFosB-ir in the GCL. The area outlined by the box is expanded below. Red arrows point to ΔFosB-labeled cells. GCL = granule cell layer. Calibration for the top row, 100 μm; for the bottom row, 50 μm.2–3. Sections from intermediate (2) and high choline (3) -treated Tg2576 mice. Same calibrations as 1. 4, A section from a WT mouse treated with the intermediate diet. Same calibrations as 1. (**B**) Quantification methods. Representative images demonstrating the thresholding criteria established to quantify ΔFosB. 1. A ΔFosB-stained section shows strongly-stained cells (white arrows). 2. A strict thresholding criterion was used to make only the darkest stained cells red. (**C**) Use of the strict threshold to quantify ΔFosB-ir. 1. Anterior dentate gyrus (DG). Tg2576 mice treated with the choline-supplemented diet had significantly less ΔFosB-ir compared to the Tg2576 mice fed the low or intermediate diets. Tg2576 mice fed the high choline diet were not significantly different from WT mice, suggesting a rescue of ΔFosB-ir. 2. There were no significant differences in ΔFosB-ir in posterior sections. (**D**) Methods are shown using a threshold that was less strict. Some of the stained cells that were included are not as dark as those used for the strict threshold (white

*Figure 7 continued on next page*

Figure 7 continued

arrows). 2. All cells above the more permissive threshold are shown in red. (**E**) Use of the less strict threshold to quantify ΔFosB-ir. 1. Anterior DG. Tg2576 mice that were fed the high choline diet had less ΔFosB-ir pixels than the mice that were fed the other diets. There were no differences from WT mice, suggesting restoration of ΔFosB-ir by choline enrichment in early life. 2. Posterior DG. There were no significant differences between Tg2576 mice fed the three diets or WT mice.

## Interictal spikes

Previous research has shown that Tg2576 mice exhibit IIS (*Figure 8A*) starting at very young ages: 4 weeks (*Kam et al., 2016*) or 6 weeks (*Bezzina et al., 2015*). Therefore, we performed video-EEG using cortical (left frontal cortex, right occipital cortex) and hippocampal (left and right) electrodes in WT and Tg2576 mice (*Figure 8B and C*). Animals were recorded for 24 hr each session so that

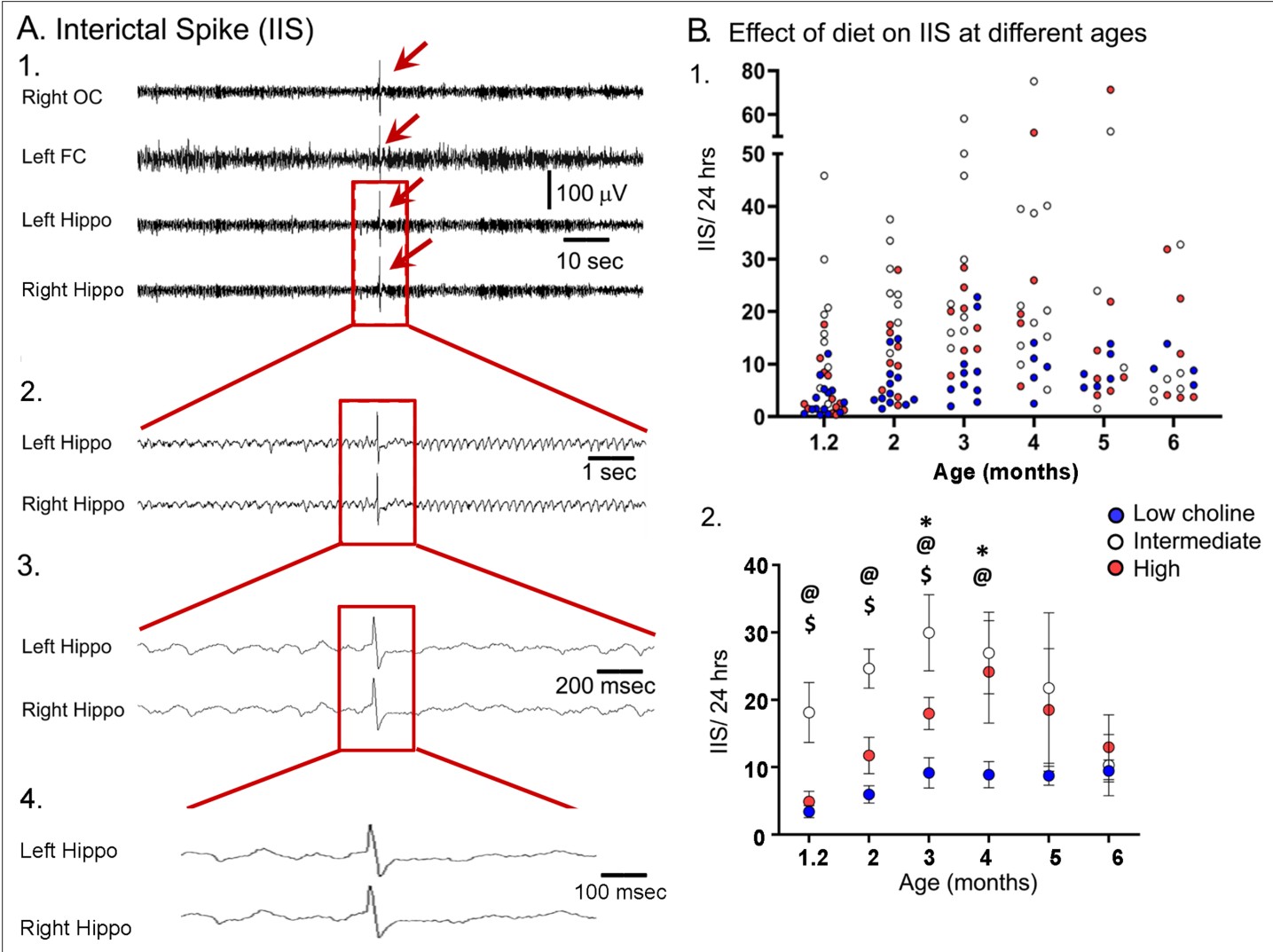

**Figure 8.** The high choline diet reduced interictal spikes (IIS) frequency in Tg2576 animals. (**A**) 1. Representative example of an IIS (red arrow). IIS were defined as occurring in all four channels (generalized) to distinguish them from focal spikes that can often be artifact. 2–4. The IIS shown in A1 is expanded. (**B**) 1. Scatter plot of IIS frequency at each age of recording. 2. Means and sem are plotted. The high choline diet group had fewer IIS than the intermediate diet group at ages 1–3 months ($, p<0.05) and the low choline group had less IIS than the intermediate diet at ages 1–4 months (@, p<0.05). The high and low choline diet groups were significantly different at ages 3 and 4 months (*, p<0.05).

The online version of this article includes the following figure supplement(s) for figure 8:

**Figure supplement 1.** Interictal spike (IIS) frequency before and after seizures.

**Figure supplement 2.** Interictal spike (IIS) frequency was similar for each sex.

the major behavioral states (exploration, awake rest, and sleep) were sampled well. Consistent with previous studies (*Bezzina et al., 2015*; *Kam et al., 2016*), we observed IIS in Tg2576 mice but not WT littermates. Therefore, analyses below were only in Tg2576 mice.

As shown in *Figure 8*, animals that received the intermediate diet had a significantly higher number of IIS in the 24 hr-long recording periods compared to animals that received the high choline and low choline diets. A two-way ANOVA (mixed model analysis) showed that there was a significant effect of age (F(2,37) = 3.38; p=0.036) and maternal diet (F(2,36) = 8.12; p=0.089). At the 5 weeks-old recording, Tukey–Kramer post hoc analyses showed that IIS frequency in animals treated with the intermediate diet was higher compared to animals treated with the low (p=0.027) or high choline diets (p=0.038). This also was true for 2 months and 3 months (low choline, high choline, p<0.05). At 4 months, the low choline group had significantly reduced IIS frequency compared to the mice that had received the intermediate diet (p=0.009) but this was not the case for the high choline group compared to the intermediate group (p=0.976). At 5–6 months, IIS frequencies were not significantly different in the mice fed the different diets (all p>0.05), probably because IIS frequency becomes increasingly variable with age (*Kam et al., 2016*). One source of variability is seizures because there was a sharp increase in IIS during the day before and after a seizure (*Figure 8—figure supplement 1*). Another reason that the diets failed to show differences was that the IIS frequency generally declined at 5–6 months. This can be appreciated in *Figure 8B* and *Figure 8—figure supplement 2*. These data are consistent with prior studies of Tg2576 mice where IIS increased from 1 to 3 months but then waxed and waned afterwards (*Kam et al., 2016*).

## Seizures and premature mortality in mice fed the low choline diet

We found that mice fed the low choline diet had greater ΔFosB-ir in GCs and the hilus showed very low NeuN-ir. Therefore, we asked whether low choline-treated mice had more seizures than mice fed the other diets. We recorded eight mice by video-EEG for 5 days each (at 6 months of age) to examine seizures and found two mice from the low choline group had seizures (11 seizures over 2 of the 5 days in one mouse, 1 seizure in the other mouse), whereas none of the other mice had seizures (n = 0/4, two intermediate and two high choline, data not shown).

These values are probably an underestimate for the low choline group because many mice in this group appeared to die in a severe seizure prior to 6 months of age (*Figure 2—figure supplement 2*). Therefore, the survivors at 6 months probably were the subset with few seizures. The reason we think that low choline-treated mice appeared to die in a seizure was that they were found in a specific posture in their cage which occurs during a severe seizure that leads to death, a prone posture with extended limbs (*Figure 2—figure supplement 2*). Regardless of how the mice died, there was greater mortality in the low choline group compared to mice that had been fed the high choline diet (log-rank [Mantel–Cox] test, chi-square 5.36, df 1, p=0.021; *Figure 2—figure supplement 2A*).

## Correlations between IIS and other measurements

As shown in *Figure 9A*, IIS were correlated to behavioral performance in some conditions. For these correlations, only mice that were fed the low and high choline diets were included because mice that were fed the intermediate diet and were tested behaviorally were not always tested with EEG.

For NOL, IIS frequency over 24 hr was plotted against the preference for the novel object in the test phase (*Figure 9A*). IIS were significantly less frequent when behavior was the best, but only for the high choline-treated mice (Pearson's r, p=0.022). In the low choline group, behavioral performance was poor regardless of IIS frequency (Pearson's r, p=0.933; *Figure 9A1*). For NOR, there were no significant correlations (low choline, p=0.202; high choline, p=0.680) but few mice were tested in the high choline-treated mice (*Figure 9B2*).

We also tested whether there were correlations between dorsal hilar NeuN-ir cell numbers and IIS frequency. In *Figure 9B*, IIS frequency over 24 hr was plotted against the number of dorsal hilar cells expressing NeuN. The dorsal hilus was used because there was no effect of diet on the posterior hilus. For NOL, there was no significant correlation (low choline, p=0.273; high choline, p=0.159; *Figure 9B1*). However, for NOR, there were more NeuN-ir hilar cells when the behavioral performance was strongest (low choline, p=0.024; high choline, p=0.016; *Figure 9B2*). These data support prior studies showing that hilar cells, especially mossy cells (the majority of hilar neurons), contribute to object recognition (*Kesner et al., 2015*; *Botterill et al., 2021*; *GoodSmith et al., 2022*).

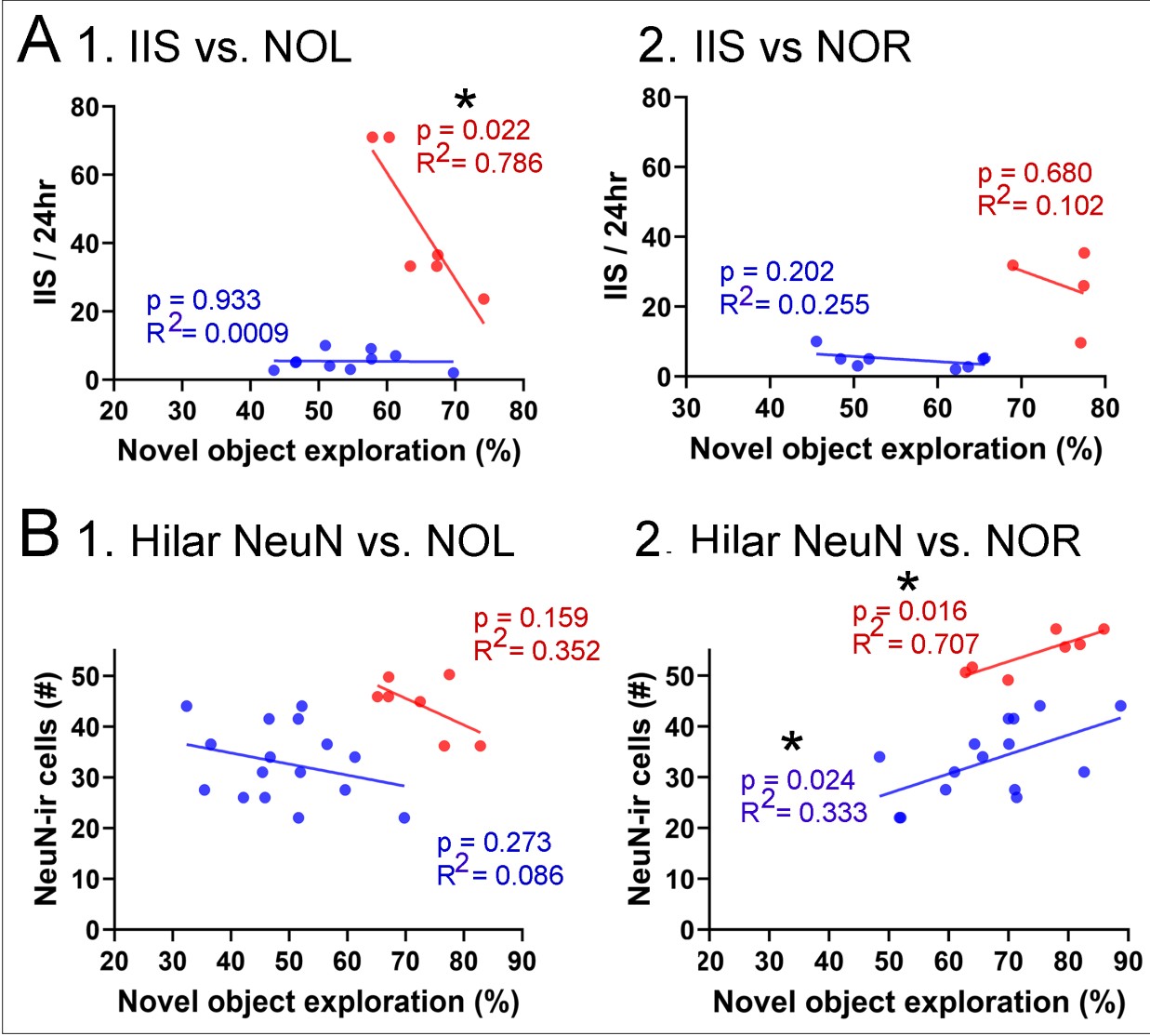

**Figure 9.** Correlations between interictal spikes (IIS), behavior, and hilar NeuN-ir. (**A**) IIS frequency over 24 hr is plotted against the preference for the novel object in the test phase of novel object location (NOL). A greater preference is reflected by a greater percentage of time exploring the novel object. 1. The mice fed the high choline diet (red) showed greater preference for the novel object when IIS were low. These data suggest IIS impaired object location memory in the high choline-treated mice. The low choline-treated mice had very weak preference and very few IIS, potentially explaining the lack of correlation in these mice. 2. There were no significant correlations for IIS and novel object recognition (NOR). However, there were only four mice for the high choline group, which is a limitation. (**B**) IIS frequency over 24 hr is plotted against the number of dorsal hilar cells expressing NeuN. The dorsal hilus was used because there was no effect of diet on the posterior hilus. 1. Hilar NeuN-ir is plotted against the preference for the novel object in the test phase of NOL. There were no significant correlations. 2. Hilar NeuN-ir was greater for mice that had better performance in NOR, both for the low choline (blue) and high choline (red) groups. These data support the idea that hilar cells contribute to object recognition (*Kesner et al., 2015*; *Botterill et al., 2021*; *GoodSmith et al., 2022*).

## Sex differences

As shown in *Figures 2 and 3*, there appeared to be no sex differences in NOL or NOR. For confirmation, we studied NOL further since that was the task that showed effects of diet. A three-way ANOVA with sex, genotype and task phase as factors showed no significant effect of sex at 3 months of age (low choline diet, $F_{(1,35)} = 0.0001$; p=0.99). Results were the same for the 3 months-old mice fed the intermediate diet ($F_{(1,22)} = 0.06$; p=0.810) and 3 months-old mice that were fed the high choline diet ($F_{(1,34)} = 0.43$; p=0.522). At 6 months of age, there also were no significant effects of sex (low choline diet, $F_{(1,23)} = 0.34$; p=0.571; intermediate diet, $F_{(1,16)} = 2.49$; p=0.130; high choline diet, $F_{(1,27)} = 0.29$; p=0.873).

Regarding IIS, we did not find sex differences (*Figure 8—figure supplement 2*). In *Figure 8—figure supplement 2A*, comparisons are made of males and females fed the low or high choline diet at 5 weeks, 2 and 3 months of age. A two-way ANOVA with sex and diet as factors showed no effect of sex (F(1,20) = 2.48; p=0.132) or diet (F(1,20) = 3.56; p=0.067). Note that for this comparison we used the low choline and high choline groups only because there were few mice for each sex at all ages. The results for months 2–3 also showed no effect of sex (2 months: F(1,17) = 0.64; p=0.429; 3 months: F(1,15) = 0.18; p=0.668).

In *Figure 8—figure supplement 2B*, we grouped mice at 1–2 months, 3–4 months, and 5–6 months so that there were sufficient females and males to compare each diet. A two-way ANOVA with diet and sex as factors showed a significant effect of diet (F(2,47) = 46.21; p<0.0001) at 1–2 months of age, but not sex (F(1,47) = 0.11, p=0.758). Post hoc comparisons showed that the low choline group had fewer IIS than the intermediate group, and the same was true for the high choline-treated mice. Thus, female mice fed the low choline diet differed from the females (p<0.0001) and males (p<0.0001) fed the intermediate diet. Male mice that had received the low choline diet different from females (p<0.0001) and males (p<0.0001) fed the intermediate diet. Female mice fed the high choline diet differed from females (p=0.002) and males (p<0.0001) fed the intermediate diet, and males fed the high choline diet differed from females (p<0.0001) and males (p<0.0001) fed the intermediate diet.

For the 3–4 months-old mice, there was also a significant effect of diet (F(2,32) = 10.82, p=0.0003) but not sex (F(1,32) = 1.05, p=0.313). Post hoc tests showed that low choline females were different from males fed the intermediate diet (p=0.007), and low choline males were also significantly different from males that had received the intermediate diet (p=0.006). There were no significant effects of diet (F(2,23) = 1.21, p=0.317) or sex (F(1,23) = 0.84, p=0.368) at 5–6 months of age.

## Discussion

### Summary

This study showed that choline supplementation in early life had several beneficial effects in Tg2576 mice. The high choline diet led to improved behavior in the NOL task, improved expression of ΔFosB, and reduced IIS frequency.

We also found surprising effects of treating mice with relatively low choline in the maternal diet. The low choline-treated Tg2576 mice had the least hilar NeuN-ir and most GCL ΔFosB protein expression. There was an impairment in the NOL task not only in low choline-treated Tg2576 mice but also WT mice. In addition, the low choline-treated Tg2576 mice showed premature mortality compared to Tg2576 mice fed the high choline diet. These results were surprising because the relatively low levels of choline have been considered by others to be normal (*Moreno et al., 2013*; *Mellott et al., 2017*; *Velazquez et al., 2019*). Our data suggest the low choline diet led to several adverse effects. Yet the IIS frequency was low in the low choline-treated Tg2576 mice, suggesting a beneficial effect. It is possible the chronic overexpression of GC ΔFosB led to reduced IIS by altering GC gene expression since ΔFosB regulates the GC transcriptome (*Corbett et al., 2017*; *You et al., 2018*; *Stephens et al., 2020*). Alterations in GC transcription could impair IIS generation because the GCs appear to be a site where IIS are generated (*Lisgaras and Scharfman, 2023*). Altered GC gene expression might also explain the reduced performance in the NOL task of low choline-treated mice because the DG plays an important role in this NOL normally (*Lee et al., 2005*).

We showed for the first time that Tg2576 mice have pathology in the hilus at an early age in that hilar neurons are deficient in immunoreactivity to anti-NeuN antibody, and this occurs before amyloid plaque develops (after 6 months of age; *Hsiao et al., 1996*). Tg2576 offspring that were fed the high choline diet showed a restoration of hilar NeuN staining. Because hilar neurons play a role in NOL (*Bui et al., 2018*), it is possible that rescue of hilar neurons by the high choline diet led to improved spatial memory. However, there was no significant correlation between the number of NeuN-ir hilar cells and testing for the NOL task. There was a significant correlation for the NOR task, with both low and high choline groups performing better when hilar NeuN-ir cell numbers were greater. The significant correlation with NOR test performance is consistent with a role of somatostatin-expressing hilar GABAergic neurons (*Nagarajan et al., 2024*) and hilar glutamatergic mossy cells in NOR and object-related activity (*Botterill et al., 2021*; *GoodSmith et al., 2022*). Somatostatin-expressing hilar neurons (*Tallent, 2007*; *Savanthrapadian et al., 2014*; *Hofmann et al., 2016*) and mossy cells

*Sloviter, 1994*; *Scharfman, 1999*; *Scharfman and Myers, 2012*; *Jinde et al., 2013*; *Bui et al., 2018*; *Botterill et al., 2019* have been suggested to contribute to excitability of GCs, so phenotypic rescue may also have contributed to the ability of the high choline diet to reduce hyperexcitability.

## Benefits of high choline

The results of this study are consistent with previous reports that prenatal or postnatal choline supplementation improves object memory. One study of iron deficiency showed that choline supplementation improved NOR (*Kennedy et al., 2014*). Wistar rats treated with a diet high (5.0 g/kg) in choline had improved NOR relative to rats that had been fed a diet lower (1.1 g/kg) in choline (*Moreno et al., 2013*). In rats that were aged to 24 months, animals that had been fed a high choline diet showed improved NOR, but only in females (*Glenn et al., 2008*).

Although few studies have examined the effects of MCS on hyperexcitability, our results showing the benefits of high choline are consistent with reports that a methyl-enriched diet (including choline but also betaine, folic acid, vitamin B12, L-methionine, and zinc) reduced spike wave discharges (*Sarkisova et al., 2023*) and audiogenic seizure severity (*Poletaeva et al., 2014*) in the offspring. Also, perinatal ethanol treatment that increased excitability was mitigated by choline chloride injection from postnatal days 10–30 (*Grafe et al., 2022*).

To our knowledge, our study is the first to show that MCS can exert effects on IIS and ΔFosB. However, prior studies have shown that MCS in rats improved memory after severe seizures. Using a convulsant to induce several hours of severe seizures (status epilepticus [SE]), there is usually memory impairment in subsequent weeks; MCS reduced the impairment (*Yang et al., 2000*; *Holmes et al., 2002*; *Wong-Goodrich et al., 2011*).

Our results are consistent with previous studies of mouse models of AD. In the APP/PS1 mouse model, where mutations found in AD are present in APP and PS1, it was shown that lifelong choline supplementation, using the high choline diet, improved memory in aged mice compared to a low choline diet (*Velazquez et al., 2019*). Postnatal choline supplementation also improved memory in APP/PS1 mice (*Wang et al., 2019*). Other improvements were also shown in the APP/PS1 mice, such as increased choline acetyltransferase, the major enyzme for acetylcholine synthesis (*Mellott et al., 2017*; *Velazquez et al., 2019*). In 3xFAD mice, where there are three familial AD mutations, a diet without choline had numerous deleterious consequences, including increased Aβ and tau phosphorylation (*Dave et al., 2023*).

## Adverse effects of the diet with relatively low choline

The relatively low choline diet had several adverse effects, which was surprising because the low choline levels are not considered very low in some prior studies (*Moreno et al., 2013*; *Mellott et al., 2017*; *Velazquez et al., 2019*). However, the past studies using low choline did not test Tg2576 mice. Also, past studies sometimes used the diet for only part of gestation rather than all of gestation and stopped after birth instead of continuing until weaning. In contrast, we fed the diet throughout gestation and until weaning. Nevertheless, it is surprising. One possible explanation is an interaction of the low choline diet with the strain of the Tg2576 mice, SJL (Swiss James Lambert). To our knowledge, this strain has not been tested with different diets before. SJL mice descend from Swiss Webster and are prone to reticulum cell sarcoma (*Haran-Ghera et al., 1967*). This strain is also characterized by vulnerability to infection causing a multiple sclerosis-like syndrome (*Linzey et al., 2023*).

One of the adverse effects was high mortality. Mice appeared to die in seizures. Mice also had high ΔFosB expression in the GCs. Another adverse effect was weak NeuN-ir in the hilus. For both WT and Tg2576 mice, NOL was impaired. These data suggest that the offspring of mothers fed the low choline diet were unhealthy. If mice were unhealthy, IIS might have been reduced due to impaired synchronization (despite high excitability in the DG). Another reason for reduced IIS is that the mice that had the low choline diet had seizures, which interrupted rapid eye movement (REM) sleep. Less REM sleep would reduce IIS because IIS occur primarily in REM (*Kam et al., 2016*). Also, seizures in the Tg2576 mice were followed by a depression of the EEG (postictal depression; *Figure 8—figure supplement 1*) that would transiently reduce IIS. A different perspective is that the intermediate diet promoted IIS (rather than low choline reducing IIS). Instead of choline, a constituent of the intermediate diet may have promoted IIS.

## NeuN

As mentioned above, NeuN is a neuronal nuclear antigen that can be phosphorylated, and when that occurs the antibody to NeuN no longer binds to NeuN. NeuN is phosphorylated in response to oxidative damage, brain injury, and toxicity (*Lind et al., 2005*). Therefore, it was of interest when we saw that hilar neurons of the DG showed reduced NeuN in Tg2576 mice. Hilar neurons are mainly glutamatergic mossy cells and somatostatin (SOM)/neuropeptide Y (NPY)-expressing GABAergic neurons (HIPP cells; *Houser, 2007*; *Scharfman and Myers, 2012*; *Scharfman, 2016*), and both neuronal types are implicated in spatial memory functions of the GCs, as well as their excitability (*Myers and Scharfman, 2009*; *Myers and Scharfman, 2011*; *Scharfman and Myers, 2012*; *Jinde et al., 2013*; *Scharfman, 2016*; *Raza et al., 2017*; *Bui et al., 2018*; *GoodSmith et al., 2019*; *Li et al., 2021*). If damaged by the high intracellular APP and Aβ levels in young Tg2576 mice, the hilar neurons would be expected to show NeuN loss. This idea is consistent with deficits in SOM and NPY-stained cells in AD (*Chan-Palay, 1987*), indicating a vulnerability.

The reason for rescue of anterior hilar but not posterior hilar NeuN with the high choline diet is unclear. The greater sensitivity of dorsal neurons may be related to differences in gene expression patterns since some of the genes that are differentially expressed along the dorsal-ventral axis could affect vulnerability to insult or injury (*Cembrowski et al., 2016*; *Zhang et al., 2018*). However, there was a similar trend for the data in both anterior and posterior regions. The lack of statistical significance in posterior DG may simply have been due to a greater variance in the posterior DG data.

## ΔFosB

Given that seizures are associated with elevated GC expression of ΔFosB (*Chen et al., 1997*; *McClung et al., 2004*; *Corbett et al., 2017*; *You et al., 2017*), the differences in ΔFosB protein expression levels observed here could be the result of increased seizures in animals that received the low choline diet. However, we did not detect a significant difference in seizures at 6 months of age, although there may have been a difference earlier in life or if we had recorded for a longer time. Interestingly, there may have been more seizures due to a reduced number of IIS. Thus, some evidence suggests that IIS can abort a seizure (*Staley and Dudek, 2006*).

ΔFosB is a transcription factor that is linked to cognition. In the J20 mouse model of AD, elevated ΔFosB in GCs led to reduced cognition, and when ΔFosB was selectively reduced the cognition improved (*Corbett et al., 2017*). Therefore, the reduction in ΔFosB by the high choline diet was important to show hyperexcitability was reduced and also important because it showed how high choline may benefit cognition.

## Choline and cholinergic neurons

There are many suggestions for the mechanisms that allow MCS to improve health of the offspring. One hypothesis that we are interested in is that MCS improves outcomes by reducing IIS. Reducing IIS would potentially reduce hyperactivity, which is significant because neuronal activity can increase release of Aβ and tau (*Cirrito et al., 2005*; *Cirrito et al., 2008*; *Bero et al., 2011*; *Yamada et al., 2014*; *Yamamoto et al., 2015*; *Hettinger et al., 2018*). IIS would also be likely to disrupt sleep since it represents aberrant synchronous activity over widespread brain regions. The disruption to sleep could impair memory consolidation since consolidation is a notable function of sleep (*Graves et al., 2001*; *Poe et al., 2010*). Indeed, in AD patients, IIS and similar events, IEDs, are correlated with memory impairment (*Vossel et al., 2016*). Sleep disruption also has other negative consequences such as impairing normal clearance of Aβ (*Nedergaard and Goldman, 2020*).

How would choline supplementation in early life reduce IIS of the offspring? It may do so by making BFCNs more resilient. That is significant because BFCN abnormalities appear to cause IIS. Thus, selective silencing of BFCNs reduced IIS (*Lisgaras and Scharfman, 2023*). The cholinergic antagonist atropine also reduces IIS when injected systemically (*Kam et al., 2016*), and it reduced elevated synaptic activity of GCs in young Tg2576 mice in vitro (*Alcantara-Gonzalez et al., 2021*). These studies are consistent with the idea that early in AD there is elevated cholinergic activity (*DeKosky et al., 2002*; *Ikonomovic et al., 2003*; *Kelley et al., 2014*; *Mufson et al., 2015*; *Kelley et al., 2016*), while later in life there is cholinergic degeneration. Indeed, the overactivity of cholinergic neurons of the first months of life could cause the degeneration at older ages.

Why would MCS make BFCNs resilient? There are several possibilities that have been explored, based on genes upregulated by MCS. One attractive hypothesis is that neurotrophic support for BFCNs is retained after MCS but after a normal diet BFCNs decline in aging and AD (*Gautier et al., 2023*). The neurotrophins, notably nerve growth factor and brain-derived neurotrophic factor, have been known for a long time to support the health of BFCNs (*Mufson et al., 2003*; *Niewiadomska et al., 2011*).

### Limitations

One of the limitations of the study was that the maternal diets were not matched exactly. Although the low and high choline diets only had differences in choline, the intermediate diet had different amounts of choline as well as other constituents (*Supplementary file 1*). Therefore, the only diet comparison with a difference restricted to choline is the low versus high choline diets (*Supplementary file 1*). The intermediate diet was useful, however, because numerous studies in AD mouse models employ this standard diet.

In addition, groups were not exactly matched. Although WT mice do not have IIS, a WT group for each of the diets would have been useful. Sample sizes were also not matched exactly because several mice died. However, this is unlikely to have had a major effect because IIS were low in frequency in all groups over the last months of the study.

Regarding sex differences, there may have been differences if females had been separated by stage of the estrous cycle at death. This possibility is raised by prior data showing that rats and mice during proestrous and estrous mornings have hyperexcitability but not at other cycle stages (*Scharfman et al., 2003*).

The Tg2576 mouse model is one of many murine models of AD, and as with all models there are inherent limitations. The Tg2576 model recapitulates familial AD, whereas the majority of AD is sporadic. Tg2576 mice also lack tau pathology. However, MCS has now demonstrated structural/functional benefits in several AD-relevant models, namely Tg2576 (this report), APP/PS1 (*Alldred et al., 2021*; *Dave et al., 2023*), Ts65Dn (*Velazquez et al., 2013*; *Powers et al., 2016*; *Strupp et al., 2016*; *Powers et al., 2017*; *Alldred et al., 2021*; *Alldred et al., 2023*), and 3xFAD mice (*Dave et al., 2023*). Moreover, increasing evidence suggests that humans with AD have low serum choline and are improved by dietary choline (*Dave et al., 2023*; *Judd et al., 2023a*; *Judd et al., 2023b*).

### Conclusions

There is now a substantial body of evidence that MCS promotes learning and memory in the offspring of normal rats and improves behavior and other abnormalities in mice that simulate AD. There also is evidence that serum levels of choline are low in AD. Therefore, it is exciting to think that dietary choline might improve AD. Given the past work also shows benefits to mice that simulate DS, the benefits of choline supplementation appear to extend to DS. The present study adds to the growing consensus that MCS is restorative by showing that hyperexcitability and pathology in the DG of Tg2576 mice are improved by MCS. They also suggest that there may be some adverse effects of a relatively low level of choline in the diet.

## Materials and methods

**Key resources table**

| Reagent type (species) or resource | Designation | Source or reference | Identifiers | Additional information |
|---|---|---|---|---|
| Antibody | Anti-NeuN (mouse monoclonal) | Cat# MAB377; Millipore | RRID:AB_2313673 | 1:5000 |
| Antibody | Anti-ΔFosB (rabbit monoclonal; D3S8R) | Cat# 14695, Cell Signaling | RRID:AB_2798577 | 1:1000 |
| Antibody | Biotinylated horse anti-mouse IgG antibody | Cat# BP-2000; Vector Laboratories | RRID:AB_2798577 | 1:500 |
| Antibody | Biotinylated goat anti-rabbit IgG antibody | Cat# BA-1000; Vector Laboratories | RRID:AB_2313606 | 1:500 |
| Chemical compound, drug | Triton-X 100 | Cat# X100; Sigma-Aldrich | | 0.25% |

*Continued on next page*

*Continued*

| Reagent type (species) or resource | Designation | Source or reference | Identifiers | Additional information |
|---|---|---|---|---|
| Chemical compound, drug | Paraformaldehyde (PFA) | Cat# 19210; Electron Microscopy Sciences | | 4% |
| Chemical compound, drug | Normal goat serum | Cat# S-1000; Vector Laboratories | | 5% |
| Chemical compound, drug | Normal horse serum | Cat# S-2000; Vector Laboratories | | 5% |
| Chemical compound, drug | TRIS hydrochloride | Cat# T3253; Sigma-Aldrich | | 97 g/8 L |
| Chemical compound, drug | Tris base (TRIZMA base) | Cat# T1503; Sigma-Aldrich | | 22 g/8 L |
| Chemical compound, drug | $H_2O_2$; hydrogen peroxide | Cat# 95321; Sigma-Aldrich | | 1% w/v |
| Chemical compound, drug | Avidin-biotin complex (ABC) | Cat# PK6100; Vector Laboratories | | 1:1000 |
| Chemical compound, drug | 3,3-Diamino-benzidine (DAB) | Cat# D5905; Sigma-Aldrich | | 0.5 mg/mL |
| Chemical compound, drug | Ammonium chloride | Cat# A514; Sigma-Aldrich | | 40 ug/mL |
| Chemical compound, drug | D(+)-glucose | Cat# G5767; Sigma-Aldrich | | 25 mg/mL |
| Chemical compound, drug | Glucose oxidase | Cat# G2133; Sigma-Aldrich | | 3 g/mL |
| Chemical compound, drug | Gelatin | Cat# G9391; Sigma-Aldrich | | 1% |
| Chemical compound, drug | Xylene | Cat# 534056; Sigma-Aldrich | | Undiluted |
| Chemical compound, drug | Permount | Cat# 17986-01; Electron Microscopy Sciences | | Undiluted |
| Chemical compound, drug | Glycerol | #G7893; Sigma-Aldrich | | 30% |
| Chemical compound, drug | Ethylene glycol | Cat# 324558; Sigma-Aldrich | | 30% |
| Chemical compound, drug | Sterile 0.9% sodium chloride solution; saline | NDC# 50989-885-17; Vedco, Inc | | |
| Chemical compound, drug | Dental cement | Cat# 4734FIB, Lang Dental Mfg. Co | | |
| Chemical compound, drug | Lactated Ringer's solution | NDC# 099355000476; Aspen Veterinary Resources Ltd | | 50 mL/kg |
| Chemical compound, drug | Isoflurane | NDC# 07-893-1389; Patterson Veterinary | | |
| Chemical compound, drug | Urethane | Cat# U2500; Sigma-Aldrich | | 2.5 g/kg |
| Chemical compound, drug | Buprenorphine | Buprenex; NDC#12496-075705; Reckitt Benckiser | | 0.2 mg/kg |
| Strain, strain background (*Mus musculus*) | Tg2576 mice, C57BL6/SJL background | Jackson Labs | Stock# 100012 | |
| Software, algorithm | ImagePro Plus V7.0 | Media Cybernetics | | |
| Software, algorithm | Sirenia acquisition | Pinnacle Technology | RRID:SCR_016183 | |
| Software, algorithm | Sirenia Seizure | Pinnacle Technology | RRID:SCR_016184 | |
| Software, algorithm | Neuroscore | Data Science International | | |
| Software, algorithm | Prism | GraphPad | RRID:SCR_002798 | |
| Software, algorithm | Photoshop | Adobe | RRID:SCR_014199 | |
| Software, algorithm | ImageJ | NIH | RRID:SCR_003070 | Version 1.44 |
| Software, algorithm | G*Power | G*Power | RRID:SCR_013726 | |
| Other | Connector | Cat# ED85100-ND; Digi-key Corporation | | For EEG |
| Other | Commutator | Cat# 8408; Pinnacle | | For EEG |

*Continued on next page*

*Continued*

| Reagent type (species) or resource | Designation | Source or reference | Identifiers | Additional information |
|---|---|---|---|---|
| Other | 0.10" stainless steel screws | Cat# 8209; Pinnacle Technology | | For EEG |
| Other | Low choline diets | AIN-76A; #110098 or #110194 Dyets, Inc | | |
| Other | High choline diet | AIN-76A; #110194 Dyets, Inc | | |
| Other | Intermediate diet | Purina 5008; W.F. Fisher and Son | | |
| Other | VIbratome | VT1000P; Leica Biosystems | | For sectioning |
| Other | Microscope slides | Cat# ZA0262; Zefon International | 3" x 1" | |
| Other | Coverslips | Cat# 48393-106 VWR Scientific Products Corp. | #1; 24 × 60 mm | |
| Other | Camera | Logitech HD Pro C920; Logitech | | For behavior |
| Other | Infrared camera | AP-DCS100W; Apex CCTV | | For EEG |
| Other | Digital camera | Model RET 2000R-F-CLR-12; Q imaging | | For microscopy |
| Other | Microscope | BX61; Olympus of America | | |
| Other | Stereotaxic apparatus | Model 902; David Kopf Instruments | | |
| Other | Peristaltic pump | Minipulse1; Gilson | | For perfusion-fixation |

## Animals

All experimental procedures followed the guidelines set by the National Institute of Health and were approved with protocol number AP2019-1640 by the Institutional Animal Care and Use Committee of the Nathan Kline Institute for Psychiatric Research (Animal Assurance number, A4545-01). Mice were housed in standard mouse cages (26 cm wide × 40 cm long × 20 cm high) with corn cob bedding and a 12 hr-long light-dark cycle. Food and water were provided ad libitum.

Tg2576 mice express a mutant form of human APP (isoform 695) with a mutation found in a Swedish family with AD (Lys670Arg, Met671Leu), driven by the hamster prion protein promoter (*Hsiao et al., 1996*). Mice were bred in-house from heterozygous Tg2576 males and non-transgenic female mice (C57BL6/SJL F1 hybrid). Wildtype (WT) mice were obtained from Jackson Laboratories (Stock# 100012). Genotypes were determined by the New York University Mouse Genotyping Core facility using a protocol to detect APP695.

Breeding pairs were randomly assigned to receive one of three diets with different concentrations of choline chloride: 1.1 g/kg (AIN-76A, Dyets Inc), 2.0 g/kg (Purina 5008, W.F. Fisher and Son Inc), and 5.0 g/kg (AIN-76A, Dyets Inc; *Supplementary file 1*). These diets ('low choline' diet, 'intermediate choline' diet, and 'high choline' diet, respectively) were used until weaning (25–30 days of age). After weaning, all mice were fed the intermediate diet. Mice were housed with others of the same sex (1–4 per cage). WT mice received the intermediate diet during breeding, gestation, and after birth.

## Behavior
### General information

Starting at least 24 hr prior to all experiments, mice were housed in the room where they would be tested behaviorally. Both NOL and NOR were preceded by three acclimation sessions (5 min each). Acclimations were conducted between 10:00 a.m. and 12:00 p.m. NOL and NOR were composed of a training and a testing session (5 min each; *Figure 1B–D*). Training and testing was conducted between 1:00 p.m. and 4:00 p.m. The interval between training and testing was 60 min, and therefore the tasks tested short-term memory (*Vogel-Ciernia and Wood, 2014*). NOL and NOR tests were separated by 7 days to ensure the effects of one task did not affect the next. Prior studies have failed to find an effect on one task on the next when a 7 day-long interval is used (*Botterill et al., 2021*). The order of testing (NOL before NOR or NOR before NOL) was randomized.

Video recordings of all training and testing sessions were captured using a USB camera (Logitech HD Pro C920, Logitech). All equipment was cleaned using 70% ethanol before each use. In pilot studies, we confirmed that objects were explored equally by animals (see 'Results'). One object was composed of red and green LEGO pieces (approximately 5 cm × 7 cm × 7 cm); the other object was a pineapple (3 cm × 3 cm × 5 cm) and made of painted metal (*Figure 1—figure supplement 1*; *Botterill et al., 2021*).

### Novel object location

NOL was conducted in a standard rat cage (26 cm wide × 40 cm long × 20 cm high) with different pictures on three sides of the cage to provide a consistent context. Pictures included several shapes and colors. The dimensions of the pictures were (1) 10 cm × 21 cm, (2) 16 cm × 18 cm, and (3) 17 cm × 20 cm. During the acclimations, animals were allowed to freely explore the cage. During the training session, mice were placed in the cage where they had been acclimated, with two identical objects, one in a corner of the cage (e.g., left top) and the other in the adjacent corner (e.g., left bottom, *Figure 1—figure supplement 1*). The mice were then removed and placed in their home cage for 1 hr. During the test session, one of the objects was moved to the opposite end of the cage (left top to right top; *Figure 1—figure supplement 1*).

### Novel object recognition

NOR was conducted in the cage described above for NOL. As for NOL, during the acclimations animals were allowed to freely explore the cage. During the training session, mice were placed in the cage used for acclimation with two identical objects centered along the shortest cage wall (*Figure 1—figure supplement 1*). The mouse was then removed and placed in their home cage for 1 hr. During the testing session, one of the objects was replaced with a different one. The objects that were identical were two pineapple-like objects, and the new object was made of red and green LEGO pieces (*Figure 1—figure supplement 1*).

### Quantification

The experimenter who conducted the analysis was blind to the genotype and which object was novel or familiar. Videos of the training and testing sessions were analyzed manually. A subset of data was analyzed by two independent blinded investigators, who were in agreement. Exploration was quantified based on the guidelines of *Vogel-Ciernia and Wood, 2014*. However, in addition to the 2014 recommendations, we also analyzed the number of approaches to an object. The time spent exploring each object was calculated as well, defined as the duration of time the nose was pointed at the object and the nose was located within 2 cm of the object. The time spent exploring also included the time animals spent on top of the object if they were looking down and sniffing it. An approach was defined by a movement toward the object and ending at least 2 cm from the edge of the object.

Animals that remembered the objects that were explored during the training phase were expected to demonstrate an increased preference for novelty in the test phase (*Ennaceur and Delacour, 1988*). In other words, exploration of the novel object during testing was expected to be higher than 50% of the total time of object exploration. Cognitive impairment was defined as significantly less than 50%.

### **Anatomy**
### Perfusion-fixation and sectioning

Mice were initially anesthetized by isoflurane inhalation (NDC# 07-893-1389, Patterson Veterinary), followed by urethane (2.5 g/kg; i.p.; Cat# U2500, Sigma-Aldrich). Under deep anesthesia, the abdominal and heart cavities were opened with surgical scissors and a 23 g needle inserted into the heart. The atria was clipped, the needle was clamped in place with a hemostat, and the animal was transcardially-perfused using 10 mL of room temperature (RT) saline (0.9% sodium chloride in double-distilled [dd] $H_2O$) using a peristaltic pump (Minipuls1; Gilson), followed by 30 mL of 4°C 4% paraformaldehyde (PFA, Cat# 19210, Electron Microscopy Sciences) in 0.1 M phosphate buffer (PB; pH 7.4). The brains were immediately removed, hemisected, and post-fixed for at least 24 hr in a scintillation vial with 10 mL 4% PFA (at 4°C).

Following post-fixation, 50-µm-thick coronal sections were made using a vibratome (VT 1000P, Leica Biosystems). Sections were collected serially and stored at 4°C in 24-well tissue culture plates containing cryoprotectant solution (30% glycerol, 30% ethylene glycol, 40% 0.1 M phosphate buffer, pH 6.7). Throughout the hippocampus, every sixth section was processed with an antibody to NeuN or ΔFosB. For analysis of anterior DG, 2–3 sections that were 300 µm apart were selected from ~1.5–2.5 mm posterior to Bregma. For posterior DG, 2–3 sections that were 300 µm apart were selected starting at ~3.5 mm posterior to Bregma. The values of the 2–3 sections were averaged so that there was one measurement for anterior and one for posterior for each mouse.

## NeuN and ΔFosB immunohistochemistry

Free-floating sections were washed in 0.1 M Tris buffer (96.96 g Tris–HCl #1185-53-1; Sigma; 22.24 g Tris Base #10708976001, q.s. to 8 L with dH$_2$O and pH to 7.6; TB; 3 × 5 min), followed by a 3 min wash in 1% (weight/volume or w/v) H$_2$O$_2$ in 0.1 M TB. Sections were then washed in 0.1 M TB (3 × 5 min) and incubated for 60 min in 5% normal horse serum for NeuN (Cat# S-2000, Vector) or 5% normal goat serum for ΔFosB (Cat# S-1000, Vector), diluted in a solution of 0.25% (volume/volume or v/v) Triton-X 100, and 1% (w/v) bovine serum albumin (#03117332001; Sigma) in 0.1 M TB. Sections were then incubated overnight at 4°C in primary antibody to NeuN (mouse monoclonal, 1:5000; Cat# MAB377, Millipore) or anti-ΔFosB (rabbit monoclonal, 1:1000; Cat# D3S8R, Cell Signaling), diluted in a solution of 0.25% (v/v) Triton-X 100, and 1% bovine serum albumin in 0.1 M TB. Both NeuN and ΔFosB have been well-characterized as antigens (*Mullen et al., 1992*; *Wolf et al., 1996*; *Chen et al., 1997*; *Sarnat et al., 1998*) and the antibodies we used have been commonly employed in the past (*Chen et al., 1997*; *Duffy et al., 2013*; *Corbett et al., 2017*). On the following day, sections were washed in 0.1 M TB (3 × 5 min) and then incubated for 60 min in biotinylated horse anti-mouse IgG secondary antibody (1:500, Cat# BP-2000, Vector) for NeuN or biotinylated goat anti-rabbit IgG secondary antibody (1:500, Cat# BA-1000) for ΔFosB, diluted in a solution of 0.25% (v/v) Triton-X 100, and 1% (w/v) bovine serum albumin in 0.1 M TB. The sections were then washed in 0.1 M TB (3 × 5 min) and incubated in avidin-biotin complex for 2 hr (1:1000; Cat# PK-6100, Vector). They were washed in 0.1 M TB (3 × 5 min) and then reacted in a solution containing 0.5 mg/mL 3, 3′-diaminobenzidine (DAB; Cat# D5905, Sigma-Aldrich), 40 µg/mL ammonium chloride (Cat# A4514, Sigma-Aldrich), 25 mg/mL D(+)-glucose (Cat# G5767, Sigma-Aldrich), and 3 g/mL glucose oxidase (Cat# G2133, Sigma-Aldrich) in 0.1 M TB. This method slowed the reaction time so that the reaction could be stopped when the immunoreactivity was robust but background was still low. The sections were then washed in 0.1 M TB (3 × 5 min), mounted on gelatin-coated (1% bovine gelatin in dH$_2$O, Cat# G9391, Sigma-Aldrich) slides (Cat# ZA0262; Zefon International) and dried at RT overnight. The following day they were dehydrated in increasing concentrations of ethanol (90%, 10 min; 95% 10 min; 100%, 10 min; 100% again, 10 min), washed in xylene (10 min; Cat# 534056, Sigma-Aldrich), and cover-slipped (Cat# 48393-106 VWR Coverglass; VWR Scientific Products Corp) with Permount (Cat# 17986-01, Electron Microscopy Sciences).

## Analysis

Photomicrographs were acquired using ImagePro Plus version 7.0 (Media Cybernetics) and a digital camera (model RET 2000R-F-CLR-12, Q-Imaging). NeuN and ΔFosB staining were quantified from micrographs using ImageJ (National Institutes of Health). All images were first converted to grayscale, and in each section, the hilus was traced, defined by zone 4 of *Amaral, 1978*. A threshold was then calculated to identify the NeuN-stained cell bodies but not background. Then NeuN-stained cell bodies in the hilus were quantified manually. Note that the threshold was defined in ImageJ using the distribution of intensities in the micrograph. A threshold was then set using a slider in the histogram provided by Image J (see *Figures 6B and 7B*). The slider was pushed from the low level of staining (similar to background) to the location where staining intensity made a sharp rise, reflecting stained cells. Cells with labeling that was above threshold were counted.

To quantify ΔFosB-stained cells, images were converted to grayscale, and in each section, the GCL was outlined and defined as a region of interest (ROI). A threshold was then set to identify the well-stained cells but not the background, analogous to the method to define threshold in the for NeuN. Two thresholds were used because some cells were lightly stained and others much more robust in their staining. The first threshold was most inclusive of immunoreactive cells. Presumably this group

reflected cells with less neuronal activity as well as those with intense activity. Then a second analysis was done with the same sections using a higher threshold. This group corresponded to cells with the highest activity. ImageJ was used to calculate the area (in pixels) within the ROI that was above threshold (see *Figure 7B*).

## Video-EEG recordings

### Stereotaxic surgery

EEG electrodes (Cat# 8209, 0.10" stainless steel screws, Pinnacle Technology) were implanted in Tg2576 animals at 4 weeks of age. WT littermates were not implanted because previous work from our laboratory has shown that WT mice have no IIS (*Kam et al., 2016*). The animals were anesthetized by isoflurane inhalation (3% isoflurane, 2% oxygen for induction) in a rectangular transparent plexiglas chamber (18 cm long × 10 cm wide × 8 cm high) made in-house. For maintenance during surgery, isoflurane was ≤1.75% and flow rate was 1.2 L/min. Mice were placed in a stereotaxic apparatus (model 902; David Kopf Instruments). Prior to the implantation of the electrodes, animals were injected with the analgesic Buprenex (0.2 mg/kg, s.c.; buprenorphine hydroxide, NDC# 12496-0757-5, Reckitt Benckiser), which was diluted in saline (0.03 mg/mL in sterile 0.9% sodium chloride solution, Vedco Inc). The skull was exposed with a midline incision and six holes were drilled for the placement of subdural screw electrodes.

The coordinates for the electrode placement were right occipital cortex (AP –3.5 mm, ML 2.0 mm), left frontal cortex (AP –0.5 mm, ML –1.5 mm), left hippocampus (AP –2.5 mm, ML –2.0 mm), and right hippocampus (AP –2.5 mm, ML 2.0 mm) (*Kam et al., 2016*). An additional screw over the right olfactory bulb was used as ground (AP 2.3 mm, ML 1.8 mm) and another screw was placed at the midline over the cerebellum (relative to Lambda: AP –1.5 mm, ML –0.5 mm) and used as reference. Screws were attached to an 8-pin connector (Cat# ED85100-ND, Digi-Key Corporation), which was placed over the skull and secured with dental cement (Cat# 4734FIB, Lang Dental Mfg. Co).

After surgery, animals were placed on a heating blanket overnight and injected with lactated Ringer's solution (50 mL/kg at 31°C; NDC# 099355000476, Aspen Veterinary Resources Ltd). They were then transferred to the room where the video-EEG was recorded.

### Video-EEG recording

Video-EEG started 1 week after surgery, at 1.25 months (5 weeks) of age. The animals were recorded at 5 weeks, and subsequently 2, 3, 4, 5, and 6 months of age. Each recording session lasted 24 hr so that a long period of sleep could be acquired, since IIS occur primarily in sleep (*Kam et al., 2016*).

Mice were placed into 21 cm × 19 cm transparent cages with food and water provided ad libitum and corncob bedding. A pre-amplifier was inserted into the 8-pin connector on the skull, which was connected to a 4-channel commutator and swivel (Cat# 8408, Pinnacle Technology). This arrangement allowed for free range of movement throughout the recording. EEG signals were acquired at 2 kHz and bandpass filtered at 0.5–200 Hz using Sirenia Acquisition (version 2.0.4, Pinnacle Technology). Simultaneous video recordings were captured using an infrared camera (Cat# AP-DCS100W, Apex CCTV).

### Analysis

EEG recordings were analyzed offline with Neuroscore version 3.2.1 (Data Science International). IIS were defined as large amplitude, 10–75 ms deflections occurring synchronously in all four leads (*Kam et al., 2016*). They were quantified by first setting an amplitude threshold (calculated separately for each recording) and a duration criterion (10–75 ms). To determine the amplitude threshold, we calculated the root mean square (RMS) amplitude for noise in each recording during a 60 s segment of baseline activity. A period of baseline activity was selected that did not include any artifacts, which were defined as extremely large deflections much greater than any other activity, and composed of a waveform that was unlike any physiological activity. They typically were accompanied by a large shake in the animal, a cable hitting a part of the cage, or scratching the implant. The threshold for IIS was set at 9 standard deviations above the RMS amplitude. This threshold was selected because it identified IIS extremely well and excluded artifacts. The IIS in one of the hippocampal channels was used instead of another channel because of data suggesting IIS begin in the hippocampus (*Kam et al., 2016*; *Lisgaras and Scharfman, 2023*). Following the automatic analysis of the IIS, the software's detection

accuracy was verified manually for each recording to ensure that all spikes appeared in all four channels and that no artifacts were included.

## Statistical comparisons

Data are expressed as mean ± standard error of the mean (SEM). The significance was set to <0.05 prior to all experiments. Tests were conducted using Prism software (GraphPad).

Parametric data comparing two groups used unpaired two-tailed $t$-tests. For >2 groups, one-way ANOVA was used followed by Tukey–Kramer post hoc tests that corrected for multiple comparisons. For data with two main factors, two-way ANOVA was followed by Tukey–Kramer post hoc tests. Interactions are not reported in the results unless they were significant. For the analysis of IIS frequency from 1.2 months of age up to 6 months of age, a repeated-measures ANOVA (RMANOVA) was used.

Tests for normality (Shapiro–Wilk) and homogeneity of variance (Bartlett's test) were used to determine whether parametric statistics could be used. When data were not normal, non-parametric data were used. When there was significant heteroscedasticity of variance, data were log transformed. If log transformation did not resolve the heteroscedasticity, non-parametric statistics were used. For non-parametric data, Mann–Whitney $U$ tests were used to compare two groups and Kruskal–Wallis for >2 groups. The post hoc tests were Dunn's test for multiple comparisons.

For correlations, Pearson's r was calculated. To compare survival curves, a log rank (Mantel–Cox) test was performed.

## Additional information

### Competing interests

Helen E Scharfman: Reviewing editor, *eLife*. The other authors declare that no competing interests exist.

### Funding

| Funder | Grant reference number | Author |
| --- | --- | --- |
| New York State Office of Mental Health | no number | Stephen D Ginsberg Helen E Scharfman |
| National Institutes of Health | AG 055328 | Helen E Scharfman Stephen D Ginsberg |
| Alzheimer's Association | AARFD-22-926807 | David Alcantara-Gonzalez |
| Natural Sciences and Engineering Research Council of Canada | RGPIN-2023-03400 | Justin J Botterill |

The funders had no role in study design, data collection and interpretation, or the decision to submit the work for publication.

### Author contributions

Elissavet Chartampila, Data curation, Formal analysis, Validation, Investigation, Visualization, Writing – original draft; Karim S Elayouby, Paige Leary, Data curation, Formal analysis, Investigation, Methodology; John J LaFrancois, Methodology; David Alcantara-Gonzalez, Justin J Botterill, Supervision, Writing – review and editing; Swati Jain, Data curation, Formal analysis, Supervision, Investigation, Writing – review and editing; Kasey Gerencer, Formal analysis, Validation, Methodology; Stephen D Ginsberg, Writing – review and editing; Helen E Scharfman, Conceptualization, Resources, Data curation, Formal analysis, Supervision, Funding acquisition, Validation, Investigation, Visualization, Methodology, Writing – original draft, Project administration, Writing – review and editing

### Author ORCIDs

Paige Leary ⓘ http://orcid.org/0000-0002-0888-466X
Helen E Scharfman ⓘ http://orcid.org/0000-0003-4006-3383

## Ethics

This study was performed in accordance with the recommendations in the Guide for the Care and Use of Laboratory Animals of the National Institutes of Health. All animals were handled according to approved institutional animal care and use committee (IACUC) protocol AP2019-649 of the Nathan Kline Institute (Animal assurance number A4545-01). All surgery was performed under deep surgical anesthesia and every effort was made to minimize animal suffering.

Joint Public Review https://doi.org/10.7554/eLife.89889.4.sa1
Author response https://doi.org/10.7554/eLife.89889.4.sa2

## Additional files

### Supplementary files

• Supplementary file 1. Comparison of the 3 diets used in this study. The major constituents of the diets are shown. Concentrations are g nutrient/kg chow. Food was provided ad libitum to breeders during mating, gestation and until weaning. After weaning all mice were fed the intermediate diet. WT mice were fed the intermediate diet during breeding, gestation, until weaning and after weaning.

### Data availability

All data are provided in Open Science Framework, a public repository. The file with the data is a project called Chartampila et al. eLife 2024, the link is: https://osf.io/u7r3k.

The following dataset was generated:

| Author(s) | Year | Dataset title | Dataset URL | Database and Identifier |
|---|---|---|---|---|
| Scharfman HE | 2024 | Chartampila et al. eLife 2024 | https://osf.io/u7r3k/ | Open Science Framework, u7r3k |

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
