## [Editor Report · eLife assessment]

In this **fundamental** work, the authors demonstrated that maternal choline supplementation improved spatial memory, reduced hyperexcitability, and restored NeuN expression in a familial Alzheimer's disease mouse model. Interestingly, choline deficiency increased mortality, while paradoxically reduced hyperexcitability. Through behavioral, electrophysiological, and histological measures, the authors present **convincing** evidence supporting the significant role of maternal choline supplementation in protecting hippocampal functions vulnerable to Alzheimer's disease.

---

## [Referee Report · Joint Public Review]

Chartampila et al. describe the effect of early-life choline supplementation on cognitive functions and epileptic activity in a mouse model of Alzheimer's disease. The cognitive abilities were assessed by the novel object recognition test and the novel object location test, performed in the same cohort of mice at 3 months and 6 months of age. Neuronal loss was tested using NeuN immunoreactivity, and neuronal hyperexcitability was examined using ΔFosB and video-EEG recordings, providing multi-level correlations between these different parameters.

The study was designed as a 6-month follow-up, with repeated behavioral and EEG measurements through disease development and multilevel correlations providing valuable and interesting findings on AD progression and the effect of early-life choline supplementation. Moreover, the behavioral data that suggest an adverse effect of low choline in WT mice are interesting and important also beyond the context of AD, highlighting the dramatic effect of diet on the phenotypes of animal.

---

## [Author Response]

The following is the authors’ response to the previous reviews.

Weaknesses:The readability could be improved.

We have gone through the paper again and tried to revise the text to improve readability.

**Reviewer #1 (Recommendations For The Authors):**
(1) Thank you for adding the discrimination ratio. However, as Fig 2 and 3 depict the same experimental data, consider harmonizing the presentation (symbols and colors) and consolidating the Figs for clarity.“

This is an excellent point but it is actually very hard to harmonize symbols and colors because the data are divided in different ways. Upon considering this further, we actually don’t want to make the symbols and colors the same because it would be misleading. For example, WT and Tg training and testing session data are divided into grey and white throughout Figure 2, but in Figure 3, training and testing session data are pooled. To color code them grey and white in Figure 3 might make it seem that in Figure 3 training and testing were separated.

(2) Fig 5 is missing

We are not sure why Figure 5 was absent since it was present in our copy of the submitted pdf. We have double checked and in the revised manuscript we are sure Figure 5 is included.

(3) Fig 6 add raw data for WT

We have added raw WT data. Revised figure 6 includes the raw data in part A4.

(4) Fig 7 add raw data for WT

We have added raw WT data. Revised Figure 7 includes the raw data in part A4.